# Message framing to promote solar panels

Dominik Bär [1] ✉, Stefan Feuerriegel [1], Ting Li[2] & Markus Weinmann [2,3]

Green technologies, such as solar panels, foster the use of clean energy, yet often involve large-scale investments. Hence, adoption by retail consumers has been a key barrier. Here, we show that message framing can significantly increase customers' serious commitment to adopting solar panels by providing empirical evidence in the field from a large-scale randomized controlled trial with a nationwide online retailer in the Netherlands ($N = 26,873$ participants). We design four messages aimed at promoting the purchase behavior of solar panel installations. Our messages present outcomes for oneself or for the environment and highlight cost savings versus earnings (for oneself) or reducing emissions versus generating green electricity (for the environment). Across all messages, we observe a higher rate of customers committing to solar panels compared to the baseline. However, the framing in terms of financial savings for oneself was by far the most effective, resulting in a 40% higher level of commitment than the baseline and 30% higher than the average of the other three messages, which were not significantly different in effect from each other. Our results show that message framing is cost-efficient and scalable among retail consumers to promote large-scale investments in green technologies and thus clean energy.

To achieve the 1.5 °C goal of the Paris Agreement, it is estimated that 70–85% of electricity generation must come from renewable energy sources by 2050[1]. This requires large-scale investments in green technologies such as solar panels[2,3]. Photovoltaic (PV) systems convert solar energy into green electricity using solar panels. In the United States, solar energy would need to account for more than 40% of total energy consumption to achieve this goal[4]. Hence, countries around the globe have implemented policies that encourage household adoption of PV systems, or hereafter simply solar panels, through financial incentives (e.g., in the United States[5], Germany[6], and the Netherlands[7]), yet such policies are costly. Here, we show that behavioral interventions, that is, interventions that do not rely on regulatory or financial incentives, are a cost-efficient alternative to promote green technologies.

Behavioral interventions have been used in various studies to promote environmentally friendly behavior[8–20]. Such interventions nudge behavioral responses in various ways, for example, by giving real-time feedback on energy consumption[9–13], setting the green option as the default[21–24] or labeling products as green[14,25–27]. Previously, behavioral interventions have been aimed at, for example,

curtailing energy consumption[9–13,28], or increasing subscriptions to green electricity tariffs[21,22,29,30]. However, research evaluating the effectiveness of behavioral interventions for large-scale investments is scarce. In this study, we thus test the effectiveness of behavioral interventions in the form of message framing in the context of large-scale investments, namely, solar panels for retail consumers. Solar panels generate green energy, which has a positive influence on various aspects of household energy consumption, such as powering heat pumps, appliances, and electric vehicles. The potential environmental impact of these investments is substantial[2,31]. Therefore, it is crucial to develop scalable and effective strategies to promote the adoption of solar panels among the general public.

Message framing refers to changes in the presentation of information, such as for products, and typically provides a scalable and cost-efficient approach to induce certain behavioral responses. As a result, message framing has been applied to various settings, such as advertising[32–34], health communication[35–37], or pro-environmental behavior[32,34,38]. In the context of pro-environmental behavior, previous research on the effect of message framing focuses on repeated or low-cost behavior[32,38–43] but not large-scale investments. For

[1]LMU Munich, Geschwister-Scholl-Platz 1, 80539 Munich, Germany. [2]Rotterdam School of Management, Erasmus University Rotterdam, Burgemeester Oudlaan 50, 3062 Rotterdam, PA, Netherlands. [3]University of Cologne, Albertus-Magnus-Platz, 50923 Cologne, Germany. ✉e-mail: baer@lmu.de

repeated or low-cost behavior such as choosing electricity tariffs, the findings are inconclusive: On the one hand, research has found that interventions targeting oneself are more effective[32,38,44]. In contrast, other studies have found interventions targeting the environment to be effective[14,39–41,45–47]. However, decision processes for repeated or low-cost behaviors are likely to be different from large-scale investment decisions, such as solar panels, as the latter requires detailed planning, significant up-front costs, and substantial time commitment[48–50]. Hence, previous findings on the effectiveness of message framing may not generalize to large-scale investments, which motivates our evaluation in the field.

There are good reasons why message framing targeting oneself or the environment might be (in)effective for promoting large-scale investments such as solar panels (see Supplementary Materials 1 for a detailed discussion). For messages targeting oneself, a common assumption is that individuals' decisions are driven by self-interest[32,41,51]. Hence, interventions highlighting economic gains, such as additional earnings or cost savings, might be more effective in motivating pro-environmental behavior. However, the large costs of solar panels mandate a thorough economic evaluation regardless of whether people are driven by self-interest such that this may not be a key determinant of the decision to adopt solar panels. In general, investment decisions may depend to a large extent on financial resources[52,53] or, at least, their perceived financial risks for an individual such that economic considerations may outweigh environmental concerns and attitudes[51,54]. People may thus be more likely to act when the decision to adopt solar panels is framed as an economic gain for oneself.

In contrast, individuals are also driven by group interests[51] and the desire to maintain a positive self-concept[39,41]. People often acknowledge the importance of environmental preservation and express a willingness to engage in environmentally friendly actions[55,56]. Therefore, pro-environmental concerns are a strong driver of human behavior[9,20,28,40,41,45–47,50]. By highlighting the environmental benefits of behaviors like preventing harm to the environment and addressing climate change, individuals may be motivated to contribute to the collective social good[51] and align with their positive self-concept[39,41]. Consequently, emphasizing the environmental gains may be effective in promoting a serious commitment to adopt solar panels.

Overall, it is thus unclear whether message framing targeted at oneself or the environment is more effective in promoting the adoption of large-scale investments in the form of solar panels. In the following, we propose messages designed to promote a serious commitment of retail consumers to adopt solar panels by targeting themselves or the environment.

For message framing targeting oneself, we test two messages, which are motivated by people's loss aversion[57]. Specifically, we highlight cost savings versus additional earnings for oneself when adopting solar panels. Framing the decision to adopt solar panels as cost savings (versus earnings) is motivated by the fact that customers may associate the term "to save" with avoiding losses (i.e., their energy bill is too high), which may cause stronger behavioral reactions compared to "to earn"[58]. In addition, framing messages as cost savings has effectively

increased support for green energy in previous research but outside of large-scale investments[44]. We thus test whether the following two messages promote solar panels: (1) *Self-Save*: "Save on average € 813 per year" and (2) *Self-Earn*: "Earn on average € 813 per year."

For message framing targeting the environment, we also test two messages, whereby we frame our message as either reducing emissions, which highlights the prevention of environmental harm or generating green electricity, which emphasizes positive environmental outcomes. Here, previous research showed that preventing a negative environmental outcome may be a stronger incentive to adopt environmentally friendly behavior but again outside of large-scale investments[59]. To this end, we study the following two interventions: (1) *Environment-CO$_2$*: "Reduce CO$_2$ emissions" and (2) *Environment-Green*: "Generate green electricity."

Overall, we test four different messages that are intentionally designed separately for each frame (see Fig. 1). As such, our messages may vary depending on the specific context of each frame so that they effectively elicit behavioral responses when targeting oneself or the environment (see Supplementary Materials 1 for details).

A shortcoming of prior research in the context of green message framing for large-scale investments is the focus on self-reported variables such as willingness to pay, attitudes, or intentions (e.g., refs. 50,60,61) while neglecting actual behavioral outcomes in the field[2,59]. Even if people report that they are willing to live environmentally friendly, it does not mean that they will act accordingly[51,62]. This observation is known as the "intention-behavior gap" and poses a severe limitation when measuring intentions instead of actual behavior. The intention-behavior gap may be especially wide for large-scale investments such as solar panels[63]. Economic abilities, regulatory frameworks, or personal living conditions can make it unfeasible to purchase solar panels but do not preclude participants in survey experiments from reporting an intent to purchase solar panels. In this study, our variable of interest differs from purchase intentions as we measure whether customers make a serious commitment to adopting solar panels in the field (i.e., whether they initiate the planning process and allocate the necessary time and resources, after having passed the feasibility check).

Solar panels have a complex sales funnel due to being high-stakes investment decisions. In contrast to many other goods (e.g., fridges, coolers, fashion), solar panels require detailed planning and custom installation, which often takes several months or even years. Thus, the first step in the sales funnel is the successful completion of a feasibility check, which is followed by a tailored planning process for installation. In our paper, we thus analyze whether a customer has made a serious commitment to invest in solar panels, i.e., they have started the planning process (and have therefore passed the feasibility check as well as have allocated the necessary time and resources for the planning process). This is a crucial step in the sales funnel and is later the dependent variable in our field experiment. Different from using completed installations, measuring commitments is beneficial in our context. The reason is that high-stakes investment decisions with custom installation naturally come with potential idiosyncrasies (e.g., some components may not be available due to production or delivery

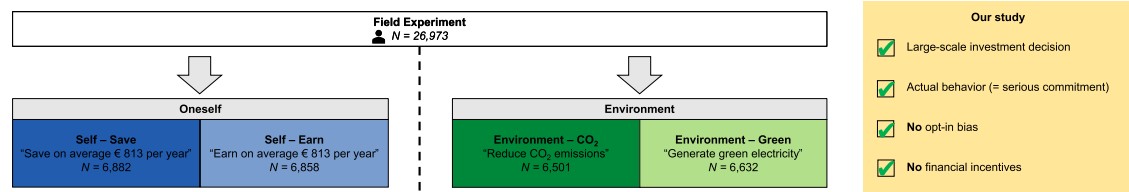

**Fig. 1 | Overview of the field experiment to study message framing.** We test the effectiveness of message framing to promote a serious commitment to adopting solar panels in a large-scale field experiment without opt-in bias or financial incentivization. Customers (*N* = 26,873) were randomized to different messages targeting either oneself or the environment.

bottlenecks or some geographic areas may also have a shortage of installation workers), and, by comparing commitments, such idiosyncrasies are avoided in our analysis.

In this work, we test the effectiveness of message framing for promoting a serious commitment to adopting solar panels in a large-scale randomized controlled trial in the field with $N = 26{,}873$ participants. Using a between-subjects design, customers visiting the website of a nationwide online retailer are randomly assigned to the four messages targeting oneself (i.e., *Self-Save*, and *Self-Earn*) or the environment (i.e., *Environment-CO₂*, and *Environment-Green*). We then estimate the causal effect of message framing through different interventions on the propensity of a customer to make a serious commitment toward adopting solar panels. Across all messages, we observe a higher rate of customers committing to solar panels compared to the baseline. However, the framing in terms of financial savings for oneself was by far the most effective, resulting in a 40% higher level of commitment than the baseline and 30% higher than the average of the other three messages, which were not significantly different in effect from each other. In our experiment, there is neither opt-in bias nor financial incentivization. Overall, our results show that message framing is cost-efficient and scalable among retail consumers to promote large-scale investments in green technologies and thus clean energy.

## Results
### Message framing promotes solar panels
To evaluate the effect of message framing for promoting solar panels, we conducted a large-scale experiment at a leading online retailer from the Netherlands. The company offers a wide range of consumer electronics but also specializes in selling and installing PV systems, which convert solar energy into green electricity using solar panels. On their website, the retailer advertised PV systems as solar panels given the widespread use of the term among potential customers but sold and installed whole PV systems (i.e., including an inverter, mounting system, etc.). Hence, we also use the term "solar panels" for simplicity. The experiment was conducted during a study period of 14 days (i.e., March 22, 2021, through April 5, 2021), during which $N = 26{,}873$ customers visited the website.

The customers visiting the website were randomly assigned to different interventions shown in a call-to-action box (see the "Methods" section). The interventions are targeted at oneself or the environment, and, for both, we test two different versions. For messages targeting oneself, we highlight cost savings or earnings. For messages targeting the environment, we frame our message as either reducing emissions or generating green electricity. All other information on the website (the product, price, layout, etc.) remained identical. Hence, differences in the percentage of customers committing to solar panels are the causal effect of our interventions. For comparison, in the two weeks before our experiment, on average, 3.8% of the customers made a serious commitment (i.e., initiating the planning process, after having passed the feasibility check) to solar panels. Website screenshots for illustration are in Supplementary Materials 3.

The effect of our different interventions is shown in Fig. 2a. We find that message framing can effectively promote solar panels. Across all messages, we observe a higher rate of customers committing to solar panels compared to the two weeks before the experiment (3.8%). However, this rate varied substantially across the different message frames. In particular, we find that the *Self-Save* frame led to the highest rate among customers (5.32%), where one in 18.7 website visitors committed to adopting solar panels. In contrast, the rate of commitment for the other message frames is only 4.17% (*Self-Earn*), 4.12% (*Environment-CO₂*), and 3.98% (*Environment-Green*). Hence, message framing that highlights cost savings for oneself can effectively promote a serious commitment to adopt solar panels.

To make statistical comparisons, we then estimated the treatment effect of the different messages on the rate of commitment among customers. The results of the logistic regression analysis are shown in Fig. 2b. We report the coefficients on the log-odds scale for reasons of comparability (the estimated rate of commitment can be obtained by applying the inverse-logit transformation $e^{\alpha_{\text{Condition}[i]}}/(1 + e^{\alpha_{\text{Condition}[i]}})$), the standard error (SE), and the 95% confidence interval (CI) in the following. Framing the decision to invest in solar panels as cost savings for oneself has the largest estimated effect (coef: −2.88, SE = 0.05, $t =$ −53.60, $P < 0.001$, 95% CI = [−2.99,−2.78]). This corresponds to an estimated rate of commitment of 5.32% for *Self-Save*. The estimated effect is thus larger than those for the other messages. This holds true for *Self-Earn* (coef: −3.13, SE = 0.06, $t = $ −51.89, $P < 0.001$, 95% CI = [−3.26,−3.02]), *Environment-CO₂* (coef: −3.15, SE = 0.06, $t = $ − 50.44, $P < 0.001$, 95% CI = [−3.27,−3.03]), and *Environment-Green* (coef: −3.18, SE = 0.06, $t = $−50.68, $P < 0.001$, 95% CI = [−3.31,−3.06]). The difference between *Self-Save* and the other messages is statistically significant, as demonstrated by 95% CIs that are non-overlapping.

As part of our robustness checks, we repeated the above regression analysis and included additional control variables as fixed effects, namely, the customers' locations, their device type (i.e., desktop, tablet, smartphone), and the weekday. This allows us to account for other forms of heterogeneity among customers. In all specifications, the findings remain robust: the condition *Self-Save* has the largest effect. Details are in Supplementary Materials 5.

In sum, framing the decision to adopt solar panels as cost savings for oneself is highly effective. As shown here, this holds true in a real-world setting for large-scale investments where message framing led to a significant increase in customers who make a serious commitment to adopt solar panels without monetary incentives.

### Environmental impact
To quantify the overall environmental impact, we conducted an environmental benefit analysis. We assumed that a household in our sample had an average annual electricity consumption of 2.81 MWh in 2021[64] and that electricity consumption of 1 MWh accounts for 0.71 t of $CO_2$ emissions[65]. Furthermore, we assume that households that make a serious commitment to solar panels eventually install solar panels. Hence, a household that covers its yearly energy consumption by installing solar panels can save 2.0 t of $CO_2$ emissions per year. By extrapolating the results from the field experiment, we can quantify the effect of our messages on $CO_2$ savings for one year (52 weeks). The condition *Self-Earn* would lead to 29,136 individuals committing to solar panels annually, amounting to annual $CO_2$ savings of 58272 t. In contrast, when using the most effective intervention (*Self-Save*), 37,171 customers would have committed to solar panels, which would save 74,342 t of annual $CO_2$ emissions. Compared to *Self-Earn*, this amounts to additional $CO_2$ savings of 16,070 t per annum (i.e., a reduction of 27.5%).

## Discussion
We performed a large-scale field experiment ($N = 26{,}873$) to assess the impact of message framing on the real-world commitment of retail consumers to adopt solar panels. We find that message framing is effective, leading to a higher rate of customers making a serious commitment to solar panels compared to the baseline rate observed in the two weeks prior to our experiment and that message framing is particularly effective when the decision to invest in solar panels was framed as cost savings for oneself ("Save on average €813 per year").

Behavioral interventions, such as message framing, have been shown to be effective in promoting behavioral change among individuals and society across various domains. Notable applications have emerged in marketing (e.g., pricing[66] or charity donations[67]) and health communication[35–37]. In the context of pro-environmental behavior, interventions have been successful in influencing repeated or low-cost behaviors, such as curtailing energy consumption[9–13,23] or increasing subscriptions to green electricity tariffs[21,22,29,30]. In contrast to these

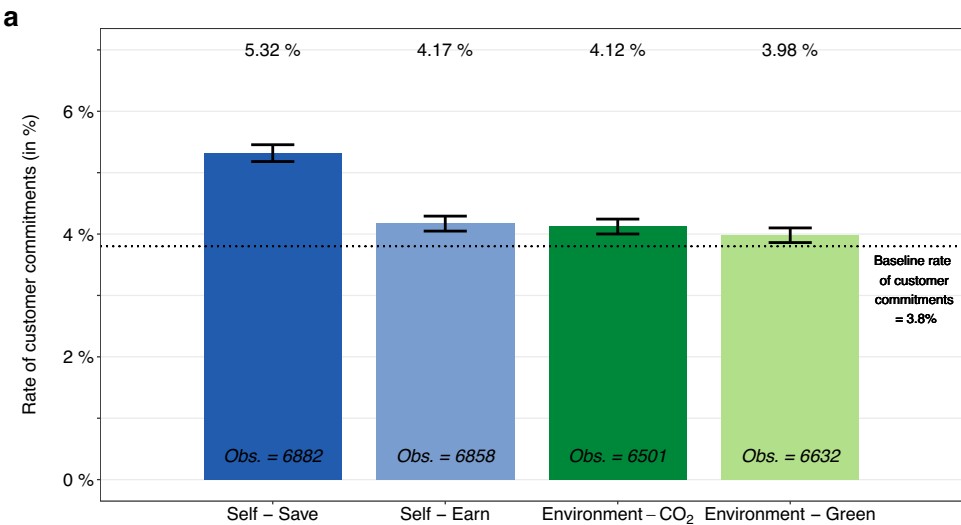

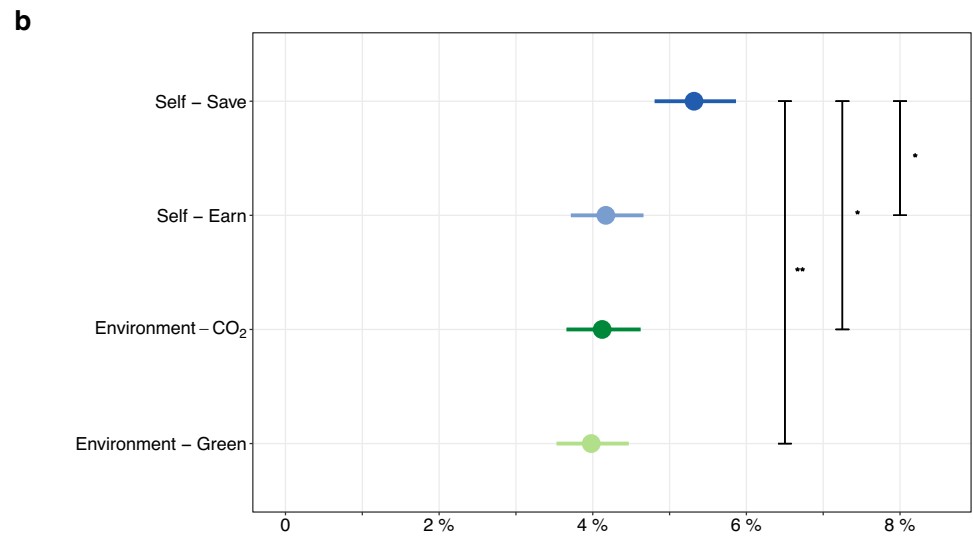

**Fig. 2 | Effect of message framing to promote solar panels in the field.**
**a** Observed rate of customers deciding to commit to solar panels and the overall number of observations by condition. Here, the observations (Obs.) are the number of customers per condition. Overall, $N = 26{,}873$ customers visited the e-commerce website during the 14-day period of the field experiment. The most effective intervention (*Self-Save*) increased the rate of customers committing to solar panels by 30% compared to the average of the other interventions. The baseline rate of commitment in the two weeks prior to the experiment was 3.8% (dotted line). Whiskers denote standard errors. **b** Estimated effect of the different interventions on the rate of customers committing to solar panels using the regression model. The coefficients capture the effect of each intervention on the decision to commit

to solar panels ($= 1$) or not ($= 0$). Across all conditions, the largest coefficient is found for *Self-Save*. The non-overlapping 95% confidence intervals (CIs) between the coefficient of *Self-Save* and the other messages imply a statistically significant treatment effect of framing solar panel investments as cost savings for oneself. The estimated coefficients are shown as the mean rate of customers committing to solar panels for each intervention (i.e., by transforming the original coefficient $\alpha_{\text{Condition[i]}}$ via $e^{\alpha_{\text{Condition[i]}}} / (1 + e^{\alpha_{\text{Condition[i]}}})$) as the mean (dot) and 95% CI (bars). Stars indicate that CIs do not overlap for 95% CIs (*) and 99% CIs (**). All statistics are based on $N = 26{,}873$ customers who visited the e-commerce website. Detailed estimation results for all coefficients are in Supplementary Materials 4.

studies, our work diverges from this existing literature by focusing on large-scale investments and demonstrates that message framing is a scalable and cost-efficient measure to promote solar panels.

As shown in our experiment, the effectiveness of message framing varies depending on whether the desired outcome targets oneself or the environment. Specifically, we observe that messages emphasizing individual benefits prove to be more effective in promoting solar panels. Given the substantial up-front costs associated with solar panels, economic considerations play a crucial role in the decision-making process[48–50,68]. Highlighting personal gains may thus reduce perceived financial risks, thereby motivating individuals to commit to solar panels. In particular, the effectiveness of framing the decisions as cost savings for oneself may be attributed to the

reason that individuals weigh perceived losses higher than perceived gains[57]. While cost savings are not strictly losses, previous literature argues that people may associate the term "to save" with avoiding losses (i.e., as their energy bill is too high)[58], thus resulting in a strong behavioral response. In the context of green energy, it was further argued that pro-environmental concerns increase the effect of interventions[9,20,39–41,45–47,50,69]. However, contrary to prior research, we do not find evidence that messages that emphasize the environmental benefits of solar panel installations are equally effective as the framing as cost savings. While the environmental aspect of solar panels may serve as an important initial motivator for customers to visit the website, it appears that the economic risks and potential gains associated with solar panels are essential for taking

the last steps and making a serious commitment to adopt solar panels.

A notable strength of our study lies in its evaluation of the causal impact of message framing for promoting green technologies in a real-world context, as we measure the serious commitment of individuals to make a large-scale investment in solar panels. Our analysis encompasses the assessment of individuals who have not only expressed serious intentions but have also taken concrete steps towards this commitment, that is, initiating the planning process and successfully completing a feasibility check. In contrast to prior literature which predominantly relies on measuring intentions rather than actual decisions[59], our research effectively bridges the intention-behavior gap, which may be particularly pronounced for large-scale investments such as solar panels. Thus, experiments in the field, as conducted in our work, are essential for assessing the effectiveness of message framing in promoting large-scale investments. Another strength of our field experiment is the ability to identify causal effects in a setting without bias due to either active opt-in or financial incentives, thereby ensuring high external validity.

Like any research, our study has certain limitations. It is important to acknowledge that customers who visit the website of the online retailer may have a higher interest in solar panels than the average public. Nevertheless, our results demonstrate that message framing can lead to a significant increase in customers' commitment to solar panels, even among individuals who may already be interested in such technologies. We also acknowledge that the wording in our messages may not be directly comparable across frames. However, this is analogous to prior research and rooted in the different motivational factors for each target (i.e., oneself or the environment). Our choice is further supported by a supplementary online experiment (see Supplementary Materials 7). Furthermore, our variable of interest measures whether customers make a serious commitment to adopting solar panels in the field. This is beneficial due to potential idiosyncrasies involved in solar panel installations which are avoided in our analysis by measuring serious commitments. In addition, our results may be specific to the setting of our field experiment, that is, solar panel installations in the Netherlands. For example, the Dutch government provides regulatory and financial incentives such as net metering or subsidies to support the installation of solar panels (see Supplementary Materials 2 for details). While similar policies are common in various countries (e.g., in the United States[5] or Germany[6]), future research may extend our analysis and study other countries or other types of green technologies where similar treatment effects may be achieved through message framing. Examples include the adoption of heat exchangers, household energy storage, or electric vehicles. In this regard, increasing the adoption of green technologies among the broader public is a policy objective in many countries[1,5–7] to which our results are thus of direct relevance.

Message framing as a non-monetary intervention provides a cost-efficient and scalable approach to promote technological change toward clean energy. Message framing may thus help mobilize private finance to increase investments in green technologies in large parts of the world. Finally, our results also provide important guidance for how policymakers should frame technology adoption when seeking to scale up clean energy, which is particularly relevant in times when the world faces the pressing challenge of the climate crisis.

## Methods

### Experimental design in the field
We evaluate the effect of message framing for promoting solar panels in a large-scale field experiment at a nationwide e-commerce company in the Netherlands. The company offers a wide assortment of consumer electronics and also sells and installs PV systems. The company advertised PV systems (i.e., including an inverter, mounting system, etc.) using the term "solar panels." Hence, we also use the term "solar

panels" for simplicity. The company is one of the leading sellers of solar panels in the country, which ensures that the results are representative. We provide a background on solar energy in the Netherlands in Supplementary Materials 2.

Our experiment follows a between-subjects design, which allows us to evaluate the separate effects of different messages on customers' likelihood to commit to solar panels. In particular, we randomly assigned customers to different messages targeting themselves or the environment.

Our messages are intentionally designed separately for each frame, taking into account prior research. As a result, there may be variations in our messages depending on the target (i.e., oneself or the environment). This approach ensures that the messages appear natural to consumers in each specific context and thus should effectively elicit behavioral responses along the dimension of oneself or the environment (see Supplementary Materials 1 for details).

For messages targeting oneself, we highlight cost savings versus earnings. Specifically, we test the following two variants: (1) *Self-Save*: "Save on average € 813 per year" and (2) *Self-Earn*: "Earn on average € 813 per year"

Both messages are designed to show concrete monetary values. This is intentional as we want to appeal to the financial motives and, thus, the immediate benefits for individuals. In fact, previous research has frequently used concrete monetary values for messages when targeting oneself[16,60,70]. By aligning with existing literature, we anticipate that individuals can easily assess their direct monetary savings or earnings. The price effect of EUR 813 includes installation and maintenance costs as well as cost savings from reduced electricity use over an amortization period of 25 years and was set by the online retailer as a conservative estimate of the potential savings that should hold for the majority of solar panels installed in the Netherlands, regardless of the specific location and other housing characteristics (see Supplementary Materials 3). We have also evaluated message variants without monetary values in a preregistered scenario-based online experiment but found that concrete values tend to result in higher adoption rates for messages targeting oneself, thus justifying our decision (see Supplementary Materials 7).

For messages targeting the environment, we highlight the prevention of environmental harm or contributions to the mitigation of climate change. In particular, we study the following two variants: (1) *Environment-CO₂*: "Reduce $CO_2$ emissions" and (2) *Environment-Green*: "Generate green electricity"

Both messages have been intentionally crafted without concrete numbers to encourage individuals to think more broadly about the collective impact on the environment. Unlike messages targeting oneself, previous research frequently uses abstract messages when targeting the environment[17,18,39,44,60,71]. This choice is driven by the challenge individuals face in quantifying environmental impacts (e.g., emissions in t $CO_2$, or energy consumption in MWh) or in relating to the corresponding numerical values[72–74]. Consequently, we adopt an abstract presentation as this is widely used in prior literature. Nevertheless, we also evaluated message variants with concrete, numerical values in a preregistered scenario-based online experiment but found that abstract messages tend to result in higher adoption rates when promoting solar panels, thus supporting our choice (see Supplementary Materials 7).

Customers visiting the website were shown one of our four messages in a prominent call-to-action box. This call-to-action box included additional information on the installation process, with a text varying according to the four messages. Importantly, the messages were always visible (even when scrolling). For each customer, the message remained the same across the entire website visit. Furthermore, through tracking, the same intervention was consistently displayed during subsequent visits throughout the entire study period, with multiple visits being captured through a single observation.

Screenshots of the website across different messages are in Supplementary Materials 3.

The experiment ran between March 22, 2021, and April 5, 2021, corresponding to a study period of 14 days. During the study period, customers arriving at the e-commerce website were randomly assigned to one of the four messages. During the experiment, there were no other simultaneous changes to the presentation, product, or price, so the differences in the adoption rates solely resulted from the different interventions. Importantly, there was neither an opt-in bias nor financial incentivization.

Ethics approval (ETH2122-0290) for the field experiment was obtained from the Rotterdam School of Management, Internal Review Board. The experimental task, data collection, and data analysis comply with all relevant ethical regulations and standard conventions in experimental economics. This ethics approval complies with regulations for studies involving human participants at the Rotterdam School of Management.

## Study population

During the study period, $N = 26,873$ customers visited the e-commerce website. This represents our study population. Customers were tracked by the website. In adherence to European Union regulations, visitors must give informed consent to tracking and personalization of ads before being able to access the website content. Thus, users without manual consent to tracking (e.g., bots, crawlers) are not included in the analysis.

For each customer, the variable of interest is whether a customer has made a serious commitment to solar panels. Note that solar panels involve a complex sales funnel that requires detailed planning followed by customized installation and construction tasks. In our analysis, the dependent variable records whether a customer has successfully completed the adoption form and thus has initiated the subsequent planning process. The completion of the adoption form necessitates considerable effort and commitment from the customer. Therefore, our variable of interest differs from a mere purchase intention, as it represents an actual decision to commit to solar panels. In particular, a customer must provide personal data and information about their property. More importantly, the adoption form includes a feasibility check informing the customer on whether it is possible to install solar panels on their property. Successful completion of the feasibility check immediately triggers the planning process and thus represents a serious commitment to solar panels.

To account for various sources of heterogeneity, we collected additional information that we later use as control variables: (i) the location (province) where the website was accessed, (ii) the device type (i.e., desktop, tablet, or mobile device), and (iii) the time of completion of the adoption form.

## Statistical analysis

To estimate the effect of message framing on customers' commitment to solar panels, we used a logistic regression model. Let the dependent variable $y$ denote whether a customer has committed to solar panels (=1) or not (=0). We then model the customer's decision of whether to commit to solar panels via

$$\log\left(\frac{P(y=1)}{1-P(y=1)}\right) = \alpha_{\text{Condition}[i]}, \tag{1}$$

where $\alpha_{\text{Condition}[i]}$ are condition-specific intercepts for the $i = 1, \ldots, 4$ conditions from above. Of note, this model specification circumvents defining a reference condition and, instead, allows us to directly estimate condition-specific mean effects. To identify the mean estimated rate of commitment of each intervention, we transform the corresponding intercept via an inverse-logit function

(e.g., $e^{\alpha_{\text{Self-Earn}}}/(1 + e^{\alpha_{\text{Self-Earn}}})$ gives the mean estimated rate of commitment for the condition *Self-Earn*). Note that coefficients may be negative on the log-odds scale, i.e., before the inverse-logit transformation. We then examine and report whether the coefficients $\alpha_{\text{Condition}[i]}$ are significantly different from zero. The statistical analysis was implemented in R 4.2.2 using the packages `stats` and `lme4`.

## Robustness checks

We conducted a series of additional robustness checks (reported in Supplementary Materials 5): (i) We repeated the analysis with alternative model specifications (i.e., probit model, linear model). (ii) We re-estimated the logistic model from the main analysis but included an intercept and then varied the reference condition. This yields four regression models, where each intervention acted once as a reference. (iii) Some customers accessed the website from outside the Netherlands, for reasons such as travel. To control for this, we re-estimated our model using only customers located in the Netherlands. (iv) We added additional control variables (i.e., device type and weekday) as fixed effects to account for other forms of heterogeneity across customers. All robustness checks led to consistent findings.

## Additional analyses

We conducted additional analyses of heterogeneity in the rate of customers committing to solar panels across rural versus urban regions. This follows the intuition that commitment to solar panels may be higher in urban areas characterized by a higher density of suitable rooftops and greater economic wealth. The analyses are in Supplementary Materials 6.

In addition to the field experiment, we conducted a scenario-based controlled online experiment ($N = 1000$) to achieve two objectives: (1) to validate the results from the field experiment, and (2) to check the comparability of our messages. The experiment was pre-registered (https://osf.io/7fnr6) and approved by the Rotterdam School of Management, Internal Review Board (ETH2122-0288). We recruited $N = 1000$ participants (mean age = 50.93 years; 54.00% women) fluent in English from the Netherlands via MSI (www.msi-aci.com). However, the majority of the Dutch population (i.e., around 90%) are fluent in English[75].

Participants first had to give informed consent and were then instructed to imagine themselves as homeowners considering the installation of solar panels. This is consistent with our field experiment where the online retailer advertised PV systems using the term "solar panels" given its widespread use among potential customers. The participants were then introduced to a fictitious e-commerce website presenting solar panels for purchase. The presentation was similar to that of the field experiment. In total, we tested eight messages targeting oneself or the environment, with outcomes presented as concrete or abstract (i.e., with or without numeric values). Compared to the field experiment, we added two abstract messages targeting oneself without numeric values and two concrete messages targeting the environment with numeric values. The reason for this is to check for comparability between messages with abstract versus concrete outcomes. The different experimental conditions are shown in Supplementary Fig. 4 of the appendix.

Using a between-subjects design, participants were asked to rate the likelihood of purchasing solar panels on this website on a Likert scale from 1 (=very unlikely) to 7 (=very likely). Consistent with the field experiment, we find that concrete messages targeting oneself tend to result in higher adoption rates for solar panels. We further find that abstract messages tend to result in higher adoption rates compared to concrete messages when targeting the environment. Hence, this motivates our choice of messages used in the field experiments. Details are in Supplementary Materials 7.

**Reporting summary**

Further information on research design is available in the Nature Portfolio Reporting Summary linked to this article.

## Data availability

All data from the online experiment is available via https://osf.io/up74v/. Because of a non-disclosure agreement, access to data for the field experiment requires authorization from the partner company. Data access can be requested via Ting Li (tli@rsm.nl). Data use agreements are subject to our partner company's availability and internal regulations.

## Code availability

All code to replicate our analyses is available via https://osf.io/up74v/.

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

## Acknowledgements

We thank our partner company for their collaboration in the implementation of the field experiment. We further thank Amyleigh Imthorn and Katharina Riepl for their help.

## Author contributions

The authors were ordered alphabetically. D.B., S.F., T.L., and M.W. contributed to conceptualization. D.B. performed the online experiments. D.B. and M.W. contributed to the data analysis. D.B., S.F., T.L., and M.W. contributed to results interpretation and manuscript writing. D.B., S.F., T.L., and M.W. approved the manuscript.

## Funding

## Competing interests

The authors declare no competing interests.
