## [Peer review file · Nature Communications]

REVIEWER COMMENTS

Reviewer #1 (Remarks to the Author):

This paper reports a field experiment on the impact of framing on consumers' adoption of PV-panels using >26000 visitors at a commercial website selling these panels in the Netherlands, randomly assigned to four different framing conditions. Supplementary analyses were done on two Prolific samples, in the USA and Netherlands. The four conditions varied in two dimensions: self- vs. environment orientations and gains vs. losses. The self-oriented loss-framing produced a substantially higher adoption rate than the other three framings. The main results were confirmed by the supplementary analyses. Most control factors had little impact in the supplementary factors, but there was a significant interaction between environmental concern and environment-loss framing, the impact of this framing being lower for those with higher environmental concern.

This is a well-designed study and a well-written and clearly reported paper. Its major strength is the field experiment, which convincingly documents the effects of different message framings on consumer adoption of PV-panels. I have a few suggestions for further improvements of an already fine manuscript.

First, I found the Methods section unnecessarily repetitive and therefore too long. It's a bit boring to re-read practically verbatim long passages from the main manuscript. It should be possible to condense the text in the Methods section more, concentrating more on the raw method.

Second, I'm surprised that we don't get any information about a baseline adoption rate. It would be interesting to know whether all the framing messages actually improved the adoption rate, but to different degree, or some had no or even a negative impact on adoption. If at all possible, the baseline adoption rate should be reported.

Third, more caution is warranted when reporting the difference between the self- and environment-directed framing. In the manuscript, this is interpreted as a self-interested drive, which is at least partly correct, but actually not very clear. There is plenty of research (back to research by Paul Stern and others in the 1990's) documenting that when it comes to bigger pro-environmental investments, economic expectations matter. However, this should be interpreted as reflecting the importance of economic constraints on families – few can ignore the economics when making bigger investments! Hence, it doesn't document that people are purely selfish maximizers, as assumed e.g., in the "economic man" metaphor.

Instead, it should be emphasized, as also mentioned in the manuscript, that this study specifically target the intention-behavior gap, which is often attributed to lacking abilities and opportunities, which includes economic abilities and opportunities (just to illustrate how old this discussion is: Ölander & Thøgersen, 1995). It actually is mute about what motivates intentions to adopt PV-panels

In this connection, it is important to consider the obvious self-selection bias in visiting a website for PV-panels. It seems likely that people who consider investing in PV-panels are more environmentally concerned than average. Their main motivation might actually be environmental. According to CLT, it is likely that they primarily thought about the abstract environmental benefits before they visited the website, and it might primarily be these reflections that made them visit the website. However, again according to CLT, as soon as they visit the website, it becomes concrete, and they start focusing more on costs and risks. If this is the case, they do not need confirmation of the environmental benefits of PV-panels. Actually, such information might very well be inconsequential. But they need comforting information that reassures them that costs and risks are at a tolerable level. This, I believe, is a more straightforward interpretation of the findings in the light of CLT, illustrating that it is unnecessary to speak of participants as driven by self-interest (which really becomes an empty statement in this framework – of course sensible people consider the economic consequences of the investment for

their family).

A small issue: The sentence lines 70-72 is difficult to follow. I would change "... along the following dimension; that is, ..." to something like: "... in terms of whether..."

Ölander, F., & Thøgersen, J. (1995). Understanding of consumer behaviour as a prerequisite for environmental protection. *Journal of Consumer Policy*, 18, 317-357.

Reviewer #2 (Remarks to the Author):

This paper clearly deserves publication. It is very unusual in the literature and helps to fill an important gap in knowledge for three reasons. (1) It is one of very few studies in the available literature that examines the effectiveness of non-financial interventions in promoting household investments that can seriously lower carbon footprints, as distinct from curtailments of energy use with technology in place (see Kastner and Stern, 2015; van der Linden, 2018). (2) It is one of only few that assesses a serious commitment to making such investments (by the way, not actual adoption, which isn't measured) as distinct from expressions of behavioral intention. And (3) it is the only study I know of that does this in a randomized control experiment, which allows for causal inferences that can be made from the correlational research that is the bulk of work in this area.

I recommend, however, that the authors make a few changes before final acceptance. One is to make clearer up front, possibly in the paragraph starting on line 29, that research on interventions with low-carbon investments is scarce.

A second recommendation, related to the first, is to avoid the common implicit assumption in work on "environmentally significant behavior" that the determinants of such behavior are the same for all of them. This assumption is evident at several places in the paper, for example, beginning with the set of citations in the paragraph starting on line 29. The available evidence suggests that the determinants of curtailments, which are most frequently studied and therefore most cited in this paper, are different from those of investments and that intrapersonal variables such as values, concerns, and attitudes have less influence on investments than on curtailments (which do not involve the significant up-front costs or careful assessment of options that are features of investments such as in photovoltaic systems) (see above citations among others). Thus, it is reasonable to expect that what is learned about curtailments will not generalize to investments. In this paper, the discussions of the literature do not always separate these classes of environmentally significant household behavior. Much is left to learn about what drives consumer investments. So, a major contribution of this paper is that it starts to fill this lacuna in available research.

I suggest that the authors be careful in their use of the term "behavioral interventions" even though it is commonly used in the field. For example, in the abstract, this phrase "behavioral interventions in the form of message framing" could be shortened to simply say "message framing", making the nature of the intervention clearer without loss of meaning. At the first use of the phrase "behavioral interventions" in the text at line 29, I suggest it be explicated, possibly this way: ", that is, interventions that do not rely on regulation or financial incentives,". All types of intervention are behavioral in that they aim to change behavior. Behavioral economics and some psychological work seeks to examine the roles and interactions of financial interventions and psychological factors in changing behavior. It is in the combination of these types of interventions where the greatest potential for mitigation seems to lie. No need to get into this in the paper. A simple explication of the term up front should do.

A similar concern is with the title. This paper looks at only one green technology: solar photovoltaics. I suggest that the title not be so expansive and simply name the technology being examined.

A minor point: In the statistical analysis in the supplementary materials, larger effects on commitment to adoption seem to be represented with negative numbers. I find this confusing. Perhaps it is explained somewhere I missed. But if not, the signs should be changed or this representation should be explained.

I have one other issue to raise. The paper presents its analysis and experimental manipulations in the frame of Construal Level Theory (CLT). This theory is unfamiliar to me from work in psychology and related fields (e.g., Nielsen et al., 2021), which builds on other theories instead. Understanding can be advanced by better communication across disciplines, literatures, and theories and this paper exemplifies lack of communication. I'm not advocating discussing other theories in this paper, but it may be worth considering ideas from other perspectives in interpreting the results here. The observed differences between the results from monetary vs. environmental framings is interpreted here in terms of the CLT categories: one framing is "closer" than the other. But there are other possibilities. They could also be interpreted as a difference between concrete and quantitative attributes (money) and abstract and qualitative ones (environment), or simply between the presence or absence of personal benefits from the investment (Wolske et al., 2017): a personal benefit of over € 800 might easily overwhelm the effects of environmental attitudes in this study. CLT doesn't offer the only reasonable explanation of the results. The results could also be discussed in yet other ways, for example, in terms of respondents' values, particularly what are called self-enhancement and self-transcendent values in Value-Belief-Norm Theory—one value type related to the importance of self enhancement and another to self transcendence (e.g., environmental and societal conditions--Wolske et al., 2017). Of course, this study can't be modified to include value measures. Also, respondents' values might not change the effectiveness of the framings used here, considering the small influence of other intrapersonal variables on household investments such as in solar energy, but the hypothesis calls out for testing. Although the paper cites some work that addresses values, do these papers investigate the relation of values to household investments? From other available evidence, I suspect that the answer may be no. I suggest that briefly addressing other possibilities raised in the literature would provide a good example of the potential of better communication across research traditions.

References

Kastner, I., & Stern, P. C. (2015). Examining the decision-making processes behind household energy investments: A review. *Energy Research & Social Science*, 10, 72–89. <http://dx.doi.org/10.1016/j.erss.2015.07.008>

Nielsen, K. S., et al., (2021). How psychology can help limit climate change. *American Psychologist*, 76(1), 130–144. <http://dx.doi.org/10.1037/amp0000624>

van der Linden, S. (2018). Warm glow is associated with low- but not high-cost sustainable behaviour. *Nature Sustainability*. 1, 28–30.

Wolske et al. (2017). Explaining interest in adopting solar photovoltaic systems in the United States: Toward an integration of behavioral theories. *Energy Research & Social Science*, 25, 134-151.

Paul C. Stern

Reviewer #3 (Remarks to the Author):

Review - Behavioral interventions increase the adoption of green technologies
NCOMMS-23-05489-T

Summary – in the paper the authors report on a large scale (N=26,873) field experiment whereby message framing was used as an intervention to test the adoption of solar panels in the Netherlands. They also describe a scenario-based experiment, which aims to describe how individual factors (especially social-demographics, social norms and environmental concern) influence the effect of framing on the adopting rate of solar panels.

Thank you for providing me with the opportunity to read and review your work. You have conducted an impressively large field experiment. However, I see fundamental theoretical and methodological issues, which is why cannot recommend publication in the current form.

1. The theoretical argument as to why the specifically message framing would have an effect is poorly developed and does not refer to the state-of-the-art literature in the field. The authors base their choice of framing on Construal Level Theory (CLT) and zoom in on the “abstract” vs “concrete” dimension of this theory, and therefore decide to focus on “self” vs “environment”. Alternative theoretical explanation (that might even fit better) are goal framing theory could be applied here or the literature on social dilemmas (short term individual vs long term collective outcomes). None of these theoretical frameworks are discussed or considered (except that there is hint of a reference to the importance of values on page 18 in the methods section, which is not where a theoretical explanation should be), and the operationalisation of the arguments is problematic (see next point). Therefore, I do not see a substantial theoretical value of this paper to support publication in comparison to previous work (e.g., Nature Climate Change volume 3, pages 413–416 (2013))

2. A 2x2 matrix to frame the messages is presented, that is consequences of adoption (for self vs environment) and type of outcome (save/ reducing vs earning/generate). There is a fundamental problem with these framing messages, because they vary on multiple dimensions, which means that it is unclear what is causing the framing effects.

a. Firstly, in the “self” framed-message, the consequences are presented in a concrete, absolute monetary values (€813) on a time scale (average per year) whereas in the “environmental” framed-message no specific details (e.g., how much CO₂? How much green energy?) without a time scale are given. This mixes up two dimensions, that is consequences for the “individual” vs the “collective” and presents these consequences in a concrete vs abstract way.

b. Secondly, in the “self” framed-message, the wording “earning” and “saving” are used whereas in the “environmental” framed-message the wording “reducing” and “generating” are used. As a result, the messages cannot be compared, and conclusions as to why they differ are not substantiated.

3. You make the bold claim that you measure actual adoption rate of solar panels and that the adoption rate of one message frame is “at least 27.50% compared to the other interventions” (abstract). I do not think the results justify these bold conclusions

a. With the given information, I seriously doubt if you, as you claim, measure behaviour instead of intentions. From your description I gather that your respondents have to take extra steps to commit to adoption solar panels on the website, that is, they have to initiate a planning process. However, it is unclear what this entails – is it an extended feasibility check? Can they withdraw? Have they signed and committed to a purchase? As it stands, the claim that you measure actual adoption rate of solar panels is not justified.

b. The main results that are presented are based on a field experiment. Respondents were people who entered the website of a retailer of solar panels. This suggests that respondents were self-selected and biased, as going to the website indicates that they had an interest in solar panels. As a result, you are overselling the impact of message framing by leaving out such nuances, e.g., in the abstract you claim: “the adoption rate increased by at least 27.50% compared to the other interventions”).

c. You draw conclusions (presented in Figure 2) on how the adoption rate of solar panels is geographically spread. The main conclusion here is that the adoption rate is higher in urban vs rural areas. There is no explanation as of why this may be the case. I also miss a discussion about the

potential influence of the study methodology on these results, e.g., internet use (and thus going to a website of a retailer that sells solar panels) might be lower in rural areas; the website of the retailer may be seen as less reliable in rural areas (perhaps they rely more on local installers?). What is the penetration rate of solar panels in the areas – could there be less interest in solar panels in rural areas, because the penetration rate is higher?

4. In addition to the main results, results of an online experiment are reported. This has merit in that it replicates the field experiment. There are pre-registered, but non-explained hypothesis on how sociodemographic, social norms and environmental concern may influence the effect of message frames on the adoption of solar panels. Conceptually, social norms and environmental concern are referred to as “behavioural attributes” which I disagree with; I’d say you look at normative assessments. In general, I see no added theoretical value of the data presented – this has been studied before, and lead to less concerns like:

a. Social norms were measured with two items?? In F2 (p53) the same item is reported twice? Assuming this is a mistake, it this is still not commonly accepted that a two-scale is used. This should at least be justified; and M, SD and reliabilities should be reported. Also what is the source of these items?

b. Environmental concerns are measured with the New Environmental Paradigm, which is a scale which has extensively been debated and many authors conclude that this is not the best scale to use to measure environmental concerns. Why is this not mentioned in the paper?

c. The survey was done among native English speakers on Academic prolific (how was this verified?) and then repeated among Dutch participants. Why not conduct this study entirely in the Netherlands, to validate the results from the field experiment. How were language issues/ translation issues dealt with? Are you merging the results of English and Dutch speakers? Where there differences? I think this is important when considering the context of the study (next point).

5. You present a field study that is conducted in the Netherlands and to interpret the results, it is essential to get a description of the context and background in this country:

a. You say in the introduction that the Netherlands provides subsidies on the installation of solar panels? How much, when, where and how and under which conditions?

b. How is the money saved/ earned done? Do owners of solar panels save money because they use their own electricity, or can they sell back to the grid? Do energy taxes play a role? What are the implications of the current energy crisis and costs on the results?

c. I don’t understand what the €813 saved/ earned refers to: is this saved in terms of electricity use? Does it consider installation costs?

I regret that I don’t have a more positive message, but I hope that this feedback is useful. I wish you all the best for the future.

RESPONSES TO REVIEWER #1

Comment R1.0: *“This paper reports a field experiment on the impact of framing on consumers’ adoption of PV-panels using >26000 visitors at a commercial website selling these panels in the Netherlands, randomly assigned to four different framing conditions. Supplementary analyses were done on two Prolific samples, in the USA and Netherlands. The four conditions varied in two dimensions: self- vs. environment orientations and gains vs. losses. The self-oriented loss-framing produced a substantially higher adoption rate than the other three framings. The main results were confirmed by the supplementary analyses. Most control factors had little impact in the supplementary factors, but there was a significant interaction between environmental concern and environment-loss framing, the impact of this framing being lower for those with higher environmental concern. This is a well-designed study and a well-written and clearly reported paper. Its major strength is the field experiment, which convincingly documents the effects of different message framings on consumer adoption of PV-panels. I have a few suggestions for further improvements of an already fine manuscript.”*

Response R1.0: Thank you for the positive summary of our submission. We gladly appreciate the detailed suggestions on how to improve our work further and, to this end, followed your feedback closely.

Comment R1.1: *“First, I found the Methods section unnecessarily repetitive and therefore too long. It’s a bit boring to re-read practically verbatim long passages from the main manuscript. It should be possible to condense the text in the Methods section more, concentrating more on the raw method”*

Response R1.1: Thank you for this comment. Following your recommendation, we have rewritten our Methods section to make it less repetitive and condensed focusing on the raw method.

Comment R1.2: *“Second, I’m surprised that we don’t get any information about a baseline adoption rate. It would be interesting to know whether all the framing messages actually improved the adoption rate, but to different degree, or some had no or even a negative impact on adoption. If at all possible, the baseline adoption rate should be reported.”*

Response R1.2: Thank you for this excellent idea. The baseline adoption rate in the two weeks prior to the experiment was 3.8%. Following your suggestion, we now state the baseline rate in our revised manuscript and compare it to our framing conditions. We have also updated Figure 2a to show the baseline rate.

Comment R1.3: *“Third, more caution is warranted when reporting the difference between the self- and environment-directed framing. In the manuscript, this is interpreted as a self-interested drive, which is at least partly correct, but actually not very clear. There is plenty of research (back to research by Paul Stern and others in the 1990’s) documenting that when it comes to bigger pro-environmental investments, economic expectations matter. However, this should be interpreted as reflecting the importance of economic constraints on families – few can ignore the economics when making bigger investments! Hence, it doesn’t document that people are purely selfish maximizers, as assumed e.g., in the “economic man” metaphor.”*

Response R1.3: Thank you for this excellent suggestion. We fully agree that economic considerations are essential for large-scale investment decisions. We have thus revised our manuscript in the following ways to address your comment:

- 1. Revised introduction:** We have revised our Introduction section to spell out more explicitly that the large costs of solar panels mandate a thorough economic evaluation regardless of whether people are driven by self-interest (as the latter may not be the only determinant of the commitment to adopt solar panels). To this end, we further discuss that investment decisions largely depend on financial resources (Gillingham et al. 2009; Ameli and Brandt 2015) and thus economic considerations may outweigh environmental concerns and attitudes (Gupta and Ogden 2009; Berger et al. 2022).
- 2. Revised discussion:** We revised the interpretation of our results in our Discussion section. We now highlight that the substantial upfront costs of solar panels make economic considerations a key factor in the decision-making process (Kastner and Stern 2015; Stern et al. 2018; van der Linden 2018). By highlighting personal gains through message framing, individuals may perceive financial risks as being lower, thereby motivating them to commit to solar panels.
- 3. Additional background.** We also included a more elaborate background in our new Appendix A called "Background".

In sum, to accommodate your suggestion, we have revised our Introduction and Discussion sections accordingly. Across our changes, we aim to offer a more accurate description of the motivation and interpretation of why message framing targeting oneself vs. the environment can promote solar panels by considering the importance of economic constraints.

Comment R1.4: *"Instead, it should be emphasized, as also mentioned in the manuscript, that this study specifically target the intention-behavior gap, which is often attributed to lacking abilities and opportunities, which includes economic abilities and opportunities (just to illustrate how old this discussion is: Ölander & Thøgersen, 1995). It actually is mute about what motivates intentions to adopt PV-panels"*

Response R1.4: Thank you for this excellent suggestion. We have made significant revisions to our manuscript to address your feedback and improve the clarity of our contribution to target the intention-behavior gap. As a result, we have revised our manuscript in the following ways:

1. In our revised Introduction section, we have added a new paragraph explicitly discussing how our study contributes to bridging the intention-behavior gap. Therein, we highlight that large-scale investments like solar panels often face significant barriers, such as economic abilities, regulatory frameworks, and personal living conditions, which may affect the feasibility of adoption but do not preclude participants in survey experiments from reporting an **intent** to purchase solar panels. However, in our study, our variable of interest differs from purchase intentions as we measure whether customers make a serious commitment to adopting solar panels in the field (i.e., whether they initiate the planning process and allocate the necessary time resources, after having passed the feasibility check).
2. We also elaborate on the above in our revised Discussion section to ensure a comprehensive and accurate explanation of our study's contributions to target the intention-behavior gap.
3. Furthermore, in response to your feedback, we have included two additional paragraphs in our revised Introduction section explaining why messages targeting oneself or the environment may be effective and what may motivate people to commit to solar panels (e.g., the desire to contribute to the collective social good, and values, norms, and beliefs). Additionally, we have added a new Appendix A (called "Background"), where we delve deeper into the underlying motivational factors.

Overall, the above revisions have significantly helped us to improve the clarity of our contribution, which is to specifically target the intention-behavior gap in the context of large-scale investments.

Comment R1.5: *“In this connection, it is important to consider the obvious self-selection bias in visiting a website for PV-panels. It seems likely that people who consider investing in PV-panels are more environmentally concerned than average. Their main motivation might actually be environmental. According to CLT, it is likely that they primarily thought about the abstract environmental benefits before they visited the website, and it might primarily be these reflections that made them visit the website. However, again according to CLT, as soon as they visit the website, it becomes concrete, and they start focusing more on costs and risks. If this is the case, they do not need confirmation of the environmental benefits of PV-panels. Actually, such information might very well be inconsequential. But they need comforting information that reassures them that costs and risks are at a tolerable level. This, I believe, is a more straightforward interpretation of the findings in the light of CLT, illustrating that it is unnecessary to speak of participants as driven by self-interest (which really becomes an empty statement in this framework – of course sensible people consider the economic consequences of the investment for their family).”*

Response R1.5: Thank you for your valuable suggestion. We appreciate your comment, and, after carefully considering your feedback and insights provided by R2 and R3, we realized that our interpretation of the results in relation to CLT was not sufficiently clear in the initial version of our manuscript. As a result, we have made significant revisions to our Discussion section. We now provide an improved interpretation along the following lines.

1. We now emphasize the significance of economic considerations in the adoption of large-scale investments, such as solar panels, and discuss the role of monetary gains in reducing perceived financial risks. We also highlight the possibility that economic motivations may simply outweigh environmental concerns.
2. We have also added a new Appendix A (called "Background") to provide further context. Therein, we elaborate on why message framing targeting oneself or the environment can be effective, providing additional information for a comprehensive understanding such as the desire to contribute to the collective social good or values, norms, and beliefs that may influence individuals' commitment to solar panels.
3. We also revised our Discussion section to acknowledge that participants who visited the website of the online retailer may have had a higher interest in solar panels compared to the average population. Our results demonstrate that message framing can still lead to a significant increase in customers' commitment to solar panels, even among individuals who may already be interested in such technologies.

Comment R1.6: *“A small issue: The sentence lines 70-72 is difficult to follow. I would change “... along the following dimension; that is, ...” to something like: “... in terms of whether...””*

Response R1.6: Thank you for this comment. We revised the sentence accordingly to make it easier to follow.

RESPONSES TO REVIEWER #2

Comment R2.0: *“This paper clearly deserves publication. It is very unusual in the literature and helps to fill an important gap in knowledge for three reasons. (1) It is one of very few studies in the available literature that examines the effectiveness of non-financial interventions in promoting household investments that can seriously lower carbon footprints, as distinct from curtailments of energy use with technology in place (see Kastner and Stern, 2015; van der Linden, 2018). (2) It is one of only few that assesses a serious commitment to making such investments (by the way, not actual adoption, which isn’t measured) as distinct from expressions of behavioral intention. And (3) it is the only study I know of that does this in a randomized control experiment, which allows for causal inferences that can be made from the correlational research that is the bulk of work in this area. I recommend, however, that the authors make a few changes before final acceptance.”*

Response R2.0: Thank you for the positive feedback and this excellent summary of our contributions. We highly appreciate the helpful recommendations to improve our work. When revising our paper, we followed your suggestions closely.

Comment R2.1: *“One is to make clearer up front, possibly in the paragraph starting on line 29, that research on interventions with low-carbon investments is scarce.”*

Response R2.1: Thank you for this comment. We have revised our Introduction section accordingly and now spell out up front that research on interventions with low-carbon investments is scarce. Overall, this helped us to distinguish our work from prior studies.

Comment R2.2: *“A second recommendation, related to the first, is to avoid the common implicit assumption in work on “environmentally significant behavior” that the determinants of such behavior are the same for all of them. This assumption is evident at several places in the paper, for example, beginning with the set of citations in the paragraph starting on line 29. The available evidence suggests that the determinants of curtailments, which are most frequently studied and therefore most cited in this paper, are different from those of investments and that intrapersonal variables such as values, concerns, and attitudes have less influence on investments than on curtailments (which do not involve the significant up-front costs or careful assessment of options that are features of investments such as in photovoltaic systems) (see above citations among others). Thus, it is reasonable to expect that what is learned about curtailments will not generalize to investments. In this paper, the discussions of the literature do not always separate these classes of environmentally significant household behavior. Much is left to learn about what drives consumer investments. So, a major contribution of this paper is that it starts to fill this lacuna in available research.”*

Response R2.2: Thank you for your insightful comment. We appreciate your feedback and the clarity it brings to the distinction between different types of environmentally significant behavior. We fully agree that large-scale pro-environmental investments, such as solar panels, involve distinct decision processes compared to repeated or low-cost behaviors.

We have followed your suggestions and have made comprehensive revisions to our manuscript as follows:

1. We have revised our Introduction and Discussion sections to explicitly separate the literature on repeated or low-cost, pro-environmentally significant behavior from the context of our study, namely, large-scale investments in the form of solar panels. To this end, we have also added the new Table 1 in our Introduction section where we categorize the existing literature according to your and R1’s feedback.

2. We now emphasize that decision processes for repeated or low-cost behaviors differ significantly from those associated with large-scale investments. We highlight that the latter requires detailed planning, substantial upfront costs, and a significant time commitment. Consequently, we emphasize that previous findings on repeated or low-cost behaviors may not necessarily generalize well to large-scale investment decisions like solar panels, as these factors introduce unique considerations.

By incorporating these revisions, our manuscript now better highlights one of our main contributions, which is to investigate the effectiveness of message framing in promoting large-scale pro-environmental behavior.

Comment R2.3: *“I suggest that the authors be careful in their use of the term “behavioral interventions” even though it is commonly used in the field. For example, in the abstract, this phrase “behavioral interventions in the form of message framing” could be shortened to simply say “message framing”, making the nature of the intervention clearer without loss of meaning. At the first use of the phrase “behavioral interventions” in the text at line 29, I suggest it be explicated, possibly this way: “, that is, interventions that do not rely on regulation or financial incentives,”. All types of intervention are behavioral in that they aim to change behavior. Behavioral economics and some psychological work seeks to examine the roles and interactions of financial interventions and psychological factors in changing behavior. It is in the combination of these types of interventions where the greatest potential for mitigation seems to lie. No need to get into this in the paper. A simple explication of the term up front should do.”*

Response R2.3: Thank you. We have thus made the following changes:

1. We shortened the terminology by replacing “behavioral interventions” with simply saying “message framing”.
2. We now explicate the term “behavioral interventions” at first use and state that we use it to refer to “interventions that do not rely on regulation or financial incentives”.
3. We have changed our title to be more specific by using “message framing” (instead of “behavioral interventions”).
4. We have also emphasized more clearly in our Introduction and Discussion sections that a salient benefit of our message framing is that it is non-monetary.

Comment R2.4: *“A similar concern is with the title. This paper looks at only one green technology: solar photovoltaics. I suggest that the title not be so expansive and simply name the technology being examined.”*

Response R2.4: Thank you for this suggestion. We changed our title to “Message framing to promote solar panels”.

Comment R2.5: *“A minor point: In the statistical analysis in the supplementary materials, larger effects on commitment to adoption seem to be represented with negative numbers. I find this confusing. Perhaps it is explained somewhere I missed. But if not, the signs should be changed or this representation should be explained.”*

Response R2.5: Thank you. We added a sentence explicating that the coefficients are in log-odds and thus may be negative before the exponential transformation. We have also added notes with explanations in our supplements to facilitate interpretation.

Comment R2.6: *“I have one other issue to raise. The paper presents its analysis and experimental manipulations in the frame of Construal Level Theory (CLT). This theory is unfamiliar to me from work in psychology and related fields (e.g., Nielsen et al., 2021), which builds on other theories instead. Understanding can be advanced by better communication across disciplines, literatures, and theories and this paper exemplifies lack of communication. I’m not advocating discussing other theories in this paper, but it may be worth considering ideas from other perspectives in interpreting the results here. The observed differences between the results from monetary vs. environmental framings is interpreted here in terms of the CLT categories: one framing is “closer” than the other. But there are other possibilities. They could also be interpreted as a difference between concrete and quantitative attributes (money) and abstract and qualitative ones (environment), or simply between the presence or absence of personal benefits from the investment (Wolske et al, 2017): a personal benefit of over € 800 might easily overwhelm the effects of environmental attitudes in this study. CLT doesn’t offer the only reasonable explanation of the results. The results could also be discussed in yet other ways, for example, in terms of respondents’ values, particularly what are called self-enhancement and self-transcendent values in Value-Belief-Norm Theory—one value type related to the importance of self enhancement and another to self transcendence (e.g., environmental and societal conditions--Wolske et al., 2017). Of course, this study can’t be modified to include value measures. Also, respondents’ values might not change the effectiveness of the framings used here, considering the small influence of other intrapersonal variables on household investments such as in solar energy, but the hypothesis calls out for testing. Although the paper cites some work that addresses values, do these papers investigate the relation of values to household investments? From other available evidence, I suspect that the answer may be no. I suggest that briefly addressing other possibilities raised in the literature would provide a good example of the potential of better communication across research traditions.”*

Response R2.6: Thank you for your insightful comment. We appreciate your suggestion to consider alternative perspectives and theories in interpreting our results, as well as the importance of communication across research traditions. After carefully considering your feedback and the insights provided by R1 and R3, we realized that our interpretation of the results in relation to CLT was not sufficiently clear in the initial version of our manuscript. We have made significant efforts to address this concern and provide alternative explanations and theoretical perspectives that could shed light on the observed differences between monetary and environmental framings.

To improve our paper, we now discuss why message framing targeting oneself versus the environment might be (in)effective for promoting solar panels. We highlight the importance of economic considerations for large-scale investments, given the substantial upfront costs and time commitment involved. Messages that target oneself by emphasizing financial benefits may be particularly effective as they reduce perceived financial risks. In contrast, we also elaborate on the individuals' desire to contribute to the collective social good and maintain a positive self-concept may motivate pro-environmental behavior.

To incorporate your feedback into our manuscript, we have made substantial revisions as follows:

- 1. Revised introduction:** We have expanded our Introduction section to provide a broader discussion on why message framing targeting oneself versus the environment might be (in)effective for promoting solar panels.
- 2. Revised discussion:** We have revised our Discussion section so that we interpret our results from different perspectives to provide a better understanding of why our conditions may be effective.
- 3. New supplementary material:** We have added a new Appendix A called "Background". Therein, we delve deeper into the scholarly discourse on whether messages targeting oneself or the environment are more effective. We elaborate on the social dilemma individuals may face when making pro-environmental decisions, discuss the role of economic considerations, and explore how individuals' values, norms, and beliefs can impact their decision-making processes when committing to solar panels.

Overall, the above revisions substantially broaden the context of why message framing may be effective in promoting solar panels. We believe these changes have strengthened our manuscript and improved the way how we spell out our contribution over the existing literature.

RESPONSES TO REVIEWER #3

Comment R3.0: *“Summary – in the paper the authors report on a large scale (N=26,873) field experiment whereby message framing was used as an intervention to test the adoption of solar panels in the Netherlands. They also describe a scenario-based experiment, which aims to describe how individual factors (especially social-demographics, social norms and environmental concern) influence the effect of framing on the adopting rate of solar panels. Thank you for providing me with the opportunity to read and review your work. You have conducted an impressively large field experiment. However, I see fundamental theoretical and methodological issues, which is why cannot recommend publication in the current form.”*

Response R3.0: Thank you for the excellent summary of our work. We gladly appreciate the detailed suggestions on how to improve our work and, to this end, followed your feedback closely.

Comment R3.1a: *“1. The theoretical argument as to why the specifically message framing would have an effect is poorly developed and does not refer to the state-of-the-art literature in the field. The authors base their choice of framing on Construal Level Theory (CLT) and zoom in on the “abstract” vs “concrete” dimension of this theory, and therefore decide to focus on “self” vs “environment”. Alternative theoretical explanation (that might even fit better) are goal framing theory could be applied here or the literature on social dilemmas (short term individual vs long term collective outcomes). None of these theoretical frameworks are discussed or considered (except that there is hint of a reference to the importance of values on page 18 in the methods section, which is not where a theoretical explanation should be), and the operationalisation of the arguments is problematic (see next point).*

Response R3.1a: Thank you for this comment, which helped us strengthen the theoretical background of our work. After reading your feedback and the insights provided by R1 and R2, we realized that our interpretation of the results in light of CLT was not sufficiently clear in the initial version of our manuscript. We have thus taken a significant effort in revising our manuscript to improve the motivation as to why message framing may be effective to promote solar panels.

We now discuss why message framing targeting oneself versus the environment might be (in)effective for promoting solar panels based on a broad set of prior literature. For example, we highlight the importance of economic considerations for large-scale investments, given the substantial upfront costs and time commitment involved as suggested by R1 and R2. In addition, we also elaborate on how individuals' desire to contribute to the collective social good and maintain a positive self-concept may motivate pro-environmental behavior (Evans et al. 2013; Bolderdijk et al. 2013).

To incorporate your feedback into our revised manuscript, we have made the following three major changes:

- 1. Revised introduction:** We have improved our Introduction section and now encompass different, explanations from prior research on why message framing targeting oneself versus the environment might be (in)effective for promoting solar panels.
- 2. Revised discussion:** We have thoroughly revised our Discussion section. We now interpret our results from different perspectives to provide a more suitable interpretation of why our interventions may be effective.
- 3. New supplementary material:** We have added a new Appendix A called "Background". Therein, we provide a broad discussion on the scholarly discourse on whether messages targeting oneself or the environment are more effective. We elaborate on the social dilemma individuals may face when making pro-environmental decisions, discuss the role of economic considerations, and explore how individuals' values, norms, and beliefs can impact their decision-making processes when committing to solar panels. We further explain how goal framing may relate to the effectiveness of our interventions.

In sum, we believe that the above changes have substantially broadened the theoretical background of our analysis as to why messages targeting oneself vs. the environment may be effective in promoting solar panels.

Comment R3.1b: *“Therefore, I do not see a substantial theoretical value of this paper to support publication in comparison to previous work (e.g., Nature Climate Change volume 3, pages 413–416 (2013))”*

Response R3.1b: Thank you for this comment and for providing us with this reference. The work by Bolderdijk et al. (2013) proved to be a highly valuable source throughout the revision process and inspired us in how we revised the theoretical background of our analysis.

We emphasize that our work differs substantially from previous work such as Bolderdijk et al. 2013. Specifically, we see three main contributions regarding how our study advances the literature on message framing for pro-environmental behavior:

- 1. Large-scale investments:** Prior work in the context of green message framing focuses on repeated or low-cost behaviors (Karp 1996; Minton and Rose 1997; Poortinga et al. 2004; Evans et al. 2013; Bolderdijk et al. 2013; Segev et al. 2015; Schwirplies and Ziegler 2016; Grazzini et al. 2018; Schwartz et al. 2020). Here, our work is unique as we focus on large-scale investments, namely, solar panels. Compared to repeated or low-cost behaviors, decision processes for large-scale investments differ significantly due to the need for detailed planning, significant up-front costs, and substantial time commitments (Kastner and Stern 2015; Rai et al. 2016; Stern et al. 2018; van der Linden 2018). Hence, previous findings on the effectiveness of message framing such as those in Bolderdijk et al. 2013 are not directly generalizable to large-scale investments. Instead, a thorough empirical evaluation in the field is mandated.
- 2. Bridging the intention-behavior gap:** As pointed out by R1, a particular shortcoming of prior research in the context of green message framing is the focus on self-reported variables such as willingness to pay, attitudes, or intentions, while neglecting actual behavioral outcomes in the field (Stern et al. 2016; Homar and Cvelbar 2021). Yet, measuring intentions instead of actual behavior is a severe limitation due to the intention-behavior gap. The intention-behavior gap may be especially wide for large-scale investments such as solar panels (Ölander and Thøgersen 1995). As such, some participants in survey experiments may report an intent, even though an actional commitment to adoption is infeasible due to the economic abilities of the participant. To address the intention-behavior gap, our variable of interest differs from purchase intentions as we measure whether customers make a serious commitment to adopting solar panels in the field (i.e., where customers initiate the planning process and allocate the necessary time resources, after having passed the feasibility check).
- 3. Non-monetary incentives:** Prior research has shown that financial incentives may crowd out pro-environmental behavior (White et al. 2019). Different from that, we focus on non-monetary behavioral interventions in the form of message framing. Hence, an important strength of our field experiment is the ability to identify causal effects in a setting without bias due to either active opt-in or financial incentives, thereby ensuring high external validity.

Throughout the revision, we realized that our contributions were not as clear as we had hoped for in our initial submission. We have thus carved out the above contributions more clearly, and, to this end, we have revised our Introduction and Discussion sections along the above lines, so that we spell out our novelty clearly. To this end, we have also revised Figure 1 and added the new Table 1 to highlight our main contributions over existing work. Furthermore, we have added a new section “Background” to our Appendix. For a detailed overview of the changes to offer an up-to-date theoretical explanation, we kindly refer to our response to Comment R3.1a.

Comment R3.2: “2. A 2x2 matrix to frame the messages is presented, that is consequences of adoption (for self vs environment) and type of outcome (save/ reducing vs earning/generate). There is a fundamental problem with these framing messages, because they vary on multiple dimensions, which means that it is unclear what is causing the framing effects.”

Response R3.2a: Thank you for your valuable feedback. To clarify our choice of how we frame messages, we improved our paper in the following ways:

1. We have clarified that we do not make comparisons across a 2x2 matrix but that we compare messages along one dimension, namely, targeting (a) oneself and (b) the environment. We intentionally compare messages along the dimension – oneself and the environment – as there is considerable uncertainty in the literature as to whether such messages are effective for promoting large-scale investments in the form of solar panels. To this end, we have rewritten our paper and spelled out that we compare messages only along one dimension. We have thus revised both our Introduction section and Fig 1.
2. We have further elaborated that we compare two variants of messages for targeting oneself vs. the environment. We state explicitly that the two variants are not designed to be directly comparable across frames. Rather, we designed the messages for each frame separately according to best practices. This is intentional, and we thus added a careful explanation to our revised manuscript (see new Introduction and Methods). We provide a short summary in our response to the next comment.

Comment R3.2a: “a. Firstly, in the “self” framed-message, the consequences are presented in a concrete, absolute monetary values (€813) on a time scale (average per year) whereas in the “environmental” framed-message no specific details (e.g., how much CO2? How much green energy?) without a time scale are given. This mixes up two dimensions, that is consequences for the “individual” vs the “collective” and presents these consequences in a concrete vs abstract way.”

Response R3.2a: Thank you. To clarify our choice of how we frame messages, we improved our paper in the following ways:

1. We have further elaborated that we compare two variants of messages for targeting oneself vs. the environment. We state explicitly that the two variants are not designed to be directly comparable across frames. Rather, we designed the messages for each frame separately according to best practices. Specifically, prior literature typically used concrete messages when targeting oneself (Poortinga and Whitaker 2018; Sussman et al. 2018; Amatulli et al. 2019), while commonly abstract messages are used for targeting the environment (Evans et al. 2013; Nabi et al. 2018; Sussman et al. 2018; Carfora et al. 2019; Ghesla et al. 2020; Gustafson et al. 2022). This is intentional, and we thus added a careful explanation to our revised manuscript (see new Introduction and Methods). We provide a short summary of our explanations in the following.

We want the messages targeting oneself to appeal to the financial motives and, thus, the immediate benefits for individuals. In fact, previous research has frequently used concrete monetary values for messages when targeting oneself (Poortinga and Whitaker 2018; Sussman et al. 2018; Amatulli et al. 2019). By aligning our messages with existing literature, we anticipate that individuals can easily assess the direct monetary impact of solar panel investments on their savings or earnings.

For messages targeting the environment, we highlight the prevention of environmental harm or contributions to the mitigation of climate change. In particular, we study the two variants “Reduce CO2 emissions” (Environment-CO2) and “Generate green electricity” (Environment-Green). Here, both messages have intentionally been crafted without concrete numbers to encourage individuals

to think more broadly about the collective impact on the environment. Unlike messages targeting oneself, previous research frequently uses abstract messages when targeting the environment (Evans et al. 2013; Nabi et al. 2018; Sussman et al. 2018; Carfora et al. 2019; Ghesla et al. 2020; Gustafson et al. 2022). This choice is driven by the challenge individuals face in quantifying environmental impacts (e.g., emissions in t CO₂, or energy consumption in MWh) or relate to the corresponding numerical values (Bleys et al. 2018; Camilleri et al. 2019; Dechezleprêtre et al. 2022). Consequently, we adopt an abstract presentation as this is widely used in prior literature.

2. We have revised our manuscript to acknowledge that our messages targeting oneself are presented with a concrete monetary value on a time scale while messages targeting the environment are abstract without mentioning a time dimension. We have further elaborated in our new Discussion section, where we acknowledge the semantic differences between the messages.
3. We validated our choice of concrete vs. abstract messages (see new Appendix G). To this end, we have conducted an additional preregistered between-subject online experiment (<https://osf.io/7fnr6>). We recruited $N=1000$ English-speaking participants from the Netherlands via MSI, a leading survey company based in the Netherlands, and repeated the online experiment from our previous submission.

Participants were instructed to imagine themselves as homeowners considering the installation of solar panels. The participants were then introduced to a fictitious e-commerce website presenting solar panels for purchase. The presentation was similar to that of the field experiment. In total, we test eight messages targeting oneself or the environment, with outcomes presented as concrete or abstract (i.e., with or without numeric values). Compared to the field experiment, we added two abstract messages targeting oneself without numeric values and two concrete messages targeting the environment with numeric values.

For messages with a concrete outcome, we test the following four conditions:

- Self-Save-Concrete: “Save on average € 813 per year”
- Self-Earn-Concrete: “Earn on average € 813 per year”
- Environment-CO₂-Concrete: “Reduce your emissions on average by 2.0 t CO₂ per year”
- Environment-Green-Concrete: “Generate on average 2.8 MWh green electricity per year”

For messages with abstract outcomes, we test the following conditions:

- Self-Save-Concrete: “Save money”
- Self-Earn-Concrete: Earn money”
- Environment-CO₂-Abstract: “Reduce CO₂ emissions”
- Environment-Green-Abstract: “Generate green electricity”

Consistent with the field experiment, the messages refer to average financial savings of € 813 per year. Furthermore, we report an environmental impact of 2.0 t CO₂ and an electricity generation of 2.8 MWh per year. These values correspond to the average CO₂ emissions and electricity consumption of a typical household in the Netherlands in 2021 (Centraal Bureau voor de Statistiek 2023a, 2023b; United States Environmental Protection Agency 2021). They represent the potential emission reduction or green electricity production achievable by installing solar panels that can cover the household's annual electricity consumption.

We find that concrete messages targeting oneself are most effective for promoting solar panels. We further find that abstract messages are more effective compared to concrete messages when targeting the environment. Hence, this motivates our choice of messages used in the field experiments. Details are in our new Appendix G titled “Online experiment”.

Overall, we believe this justifies the choice of our messages and strengthens the validity of our findings.

Comment R3.2b: *“b. Secondly, in the “self” framed-message, the wording “earning” and “saving” are used whereas in the “environmental” framed-message the wording “reducing” and “generating” are used. As a result, the messages cannot be compared, and conclusions as to why they differ are not substantiated.”*

Response R3.2b: Thank you for bringing up this important point. We appreciate the opportunity to clarify our rationale behind the choice of wording.

The intention behind using different wording in the messages targeting oneself and the environment was to align with the motivations emphasized in each context. The "self" framed messages emphasize financial benefits, such as cost savings or additional earnings, while the "environment" framed messages focus on environmental outcomes, such as reducing environmental harm or generating positive environmental impact. We further use terminology that would be most familiar to customers in each respective context. For instance, financial benefits are commonly associated with terms like "savings" or "earnings," while environmental outcomes are often described using terms like "reduce" or "generate" concerning CO2 emissions or green energy. We have clarified our rationale for designing the messages in our revised Introduction section.

Our above choice was further motivated by our aim to compare message framing along one dimension: “self” vs “environment”. Hence, we designed the variants for each frame according to best-practice in the literature (Evans et al. 2013; Nabi et al. 2018; Sussman et al. 2018; Carfora et al. 2019; Ghesla et al. 2020; Gustafson et al. 2022; Poortinga and Whitaker 2018; Amatulli et al. 2019). Because of this, we also avoid direct comparisons about specific terms (e.g., comparing “earning” from self vs. “generating” from the environment). We have further elaborated in our new Discussion section, where we acknowledge the semantic differences between the messages. However, in our study, the implied meaning of the wording largely depends on the context of the message, either emphasizing oneself or the environment. Therefore, we believe it is crucial to use wording that is contextually appropriate and familiar to ensure the validity of our findings. In that sense, our study is in line with prior literature analyzing car sharing (Evans et al. 2013), the intentions to adopt electric vehicles (Herberz et al. 2022), and support for renewable energy (Gustafson et al. 2022) but where conditions with different wordings are used as in our study.

To strengthen our work further and to thoroughly address your comment, we have thus discussed your comment as part of our limitations (see revised Discussion section).

Comment R3.3a: *“3. You make the bold claim that you measure actual adoption rate of solar panels and that the adoption rate of one message frame is “at least 27.50% compared to the other interventions” (abstract). I do not think the results justify these bold conclusions*

a. With the given information, I seriously doubt if you, as you claim, measure behaviour instead of intentions. From your description I gather that your respondents have to take extra steps to commit to adoption solar panels on the website, that is, they have to initiate a planning process. However, it is unclear what this entails – is it an extended feasibility check? Can they withdraw? Have they signed and committed to a purchase? As it stands, the claim that you measure actual adoption rate of solar panels is not justified.”

Response R3.3a: Thank you for this comment. After carefully considering your feedback, we have made significant revisions to our manuscript to address your concerns regarding the measurement of our dependent variable:

1. We agree that the wording in our initial submission may have been not as clear as we had hoped for. We have thus revised our wording by following the suggestions from you and R2. As a result, we replaced the term “adoption” with “(serious) commitment.” This should accurately describe the meaning of our dependent variable and clarify its interpretation.
2. We realized that more information is necessary to describe our dependent variable. We have thus added additional explanations to our Introduction section and our Method section. Therein, we

explain that customers who make a serious commitment to solar panels (i) have successfully completed a feasibility check to evaluate whether their property qualifies for the installation of solar panels and (ii) initiated the planning process for installation. As a result of this, customers have submitted detailed personal information and thus make a serious time commitment to planning and thus towards adopting solar panels in a real-world setting.

3. We further added a justification to our Introduction and Discussion sections as to why our choice of the dependent variable is beneficial for our setting involving large-scale investments. The reason is that solar panels are large-scale investments that require custom planning and custom installation. However, such custom installation naturally comes with several idiosyncrasies that are inherent to the complex and lengthy sales funnel for solar panels (e.g., some components may not be available due to production or delivery bottlenecks, or some geographic areas may also have a shortage of installation workers). We acknowledge that such idiosyncrasies may lead to dropout, while, by measuring and analyzing commitments, we can avoid them in our analysis. Likewise, using completed installations is further non-trivial or even infeasible simply due to the length of the sales funnel as installation may take several months or even years. To address your comment, we discuss our choice of the dependent variable in our Discussion and acknowledge potential limitations.
4. We have added further context from the literature on the intention-behavior gap, which allows us to explain how our work is unique (see our revised Introduction, our revised Methods, and our new Appendix A). To this end, we spell out explicitly why our dependent variable is different from intentions. In survey experiments where intents are measured, customers may state an interest in solar panels even when their economic abilities, personal living, etc. may not allow them to do so. In contrast to that, we measure a serious commitment to solar panels where participants actively decide to commit to solar panels in the field by initiating the planning process and by allocating the necessary time resources, after having passed the feasibility check.

In sum, we greatly appreciate your valuable input, which has contributed to the refinement of our manuscript.

Comment R3.3b: *“b. The main results that are presented are based on a field experiment. Respondents were people who entered the website of a retailer of solar panels. This suggests that respondents were self-selected and biased, as going to the website indicates that they had an interest in solar panels. As a result, you are overselling the impact of message framing by leaving out such nuances, e.g., in the abstract you claim: “the adoption rate increased by at least 27.50% compared to the other interventions”).”*

Response R3.3b: Thank you for your comment. As a result, we have revised our manuscript in the following ways:

1. We have revised our manuscript and adopted a more reserved tone when reporting our results. We acknowledge that the respondents in our field experiment may have had a prior interest in installing solar panels, as indicated by their visit to the retailer's website. We have further clarified that prior interest in solar panels in our sample may influence our results.
2. We have extended our Discussion section. Therein, we now discuss that self-selection could introduce bias and potentially impact the generalizability of our findings to the broader public. Nevertheless, we also state clearly that message framing can still lead to a significant increase in customers' commitment to solar panels, *even* among individuals who may already be interested in such technologies.
3. We added the baseline rate of users committing to solar panels before the experiment (see new materials in the Results section). We thereby make it easier for readers to assess and interpret the effectiveness of message framing.
4. We strengthened the motivation for our online experiment. Therein, we elaborate that our online experiment does not have such a bias toward individuals who are already interested in such

technologies. Our online experiment thus provides a validation (where the results are in line with the field experiment).

Comment R3.3c: *“c. You draw conclusions (presented in Figure 2) on how the adoption rate of solar panels is geographically spread. The main conclusion here is that the adoption rate is higher in urban vs rural areas. There is no explanation as of why this may be the case. I also miss a discussion about the potential influence of the study methodology on these results, e.g., internet use (and thus going to a website of a retailer that sells solar panels) might be lower in rural areas; the website of the retailer may be seen as less reliable in rural areas (perhaps they rely more on local installers?). What is the penetration rate of solar panels in the areas – could there be less interest in solar panels in rural areas, because the penetration rate is higher?”*

Response R3.3c: Thank you for bringing up this important point regarding the analysis of regional heterogeneity in the commitment to solar panels. We realized that our previous presentation of the results was not as clear as intended. Hence, we added a potential explanation that explains the variation (see Appendix F). Therein, we discuss factors such as penetration rates of solar panels and internet use. After reading your comments and those from the other reviewers, we realized that we should clearly position our manuscript around our main contribution, that is, the field experiment, and we thus find it beneficial to move the above analysis to the supplements.

Comment R3.4a: *“4. In addition to the main results, results of an online experiment are reported. This has merit in that it replicates the field experiment. There are pre-registered, but non-explained hypothesis on how sociodemographic, social norms and environmental concern may influence the effect of message frames on the adoption of solar panels. Conceptually, social norms and environmental concern are referred to as “behavioural attributes” which I disagree with; I’d say you look at normative assessments. In general, I see no added theoretical value of the data presented – this has been studied before, and lead to less concerns like:*

Response R3.4a: Thank you for this feedback. We improved our paper in two ways:

1. We strengthened the focus around our main contribution. After reading your comments, we realized that the additional analyses around sociodemographics, social norms, and environmental concerns are of little value. Hence, following feedback from you and the other reviewers, we decided to reposition our manuscript around our main contribution, that is, the field experiment. As a result, we have removed the additional analyses.
2. We still report the results from the online experiment (but without the additional analyses) as it allows us to replicate the field experiment. We further moved the online experiment from the main paper to the supplements to strengthen our focus on the field experiment.

Comment R3.4b: *“a. Social norms were measured with two items?? In F2 (p53) the same item is reported twice? Assuming this is a mistake, it this is still not commonly accepted that a two-scale is used. This should at least be justified; and M, SD and reliabilities should be reported. Also what is the source of these items?”*

Response R3.4b: Thank you for this comment. Following feedback from you and the reviewer team, we realized that our additional analysis based on social norms was of little novelty. We have thus removed them from our manuscript. This allows us to shift the focus of our manuscript toward our main contribution, that is, the field experiment.

Comment R3.4c: *“b. Environmental concerns are measured with the New Environmental Paradigm, which is a scale which has extensively been debated and many authors conclude that this is not the best scale to use to measure environmental concerns. Why is this not mentioned in the paper?”*

Response R3.4c: Thank you for this comment. In line with the above comment from you and the feedback from the entire reviewer team, we found that the additional analysis around environmental concerns was of little overall value. Hence, we have removed the analysis as well as the corresponding scale.

Comment R3.4d: *“c. The survey was done among native English speakers on Academic proflific (how was this verified?) and then repeated among Dutch participants. Why not conduct this study entirely in the Netherlands, to validate the results from the field experiment. How were language issues/ translation issues dealt with? Are you merging the results of English and Dutch speakers? Where there differences? I think this is important when considering the context of the study (next point).”*

Response R3.4d: Thank you for your feedback. We have revised our online experiment (see our revised Appendix G). Specifically, we have updated our online experiment with Dutch participants. As a result, we are now consistent with our field experiment, so that we can validate the results from the latter. To ensure effective communication and data collection as well as reproducibility for follow-up studies in other parts of the world, our experiment was conducted in English, and therefore, we specifically recruited participants who are fluent in English. However, the majority of the Dutch population (i.e., around 90%) are fluent in English (European Commission 2012).

Comment R3.5a: *“5. You present a field study that is conducted in the Netherlands and to interpret the results, it is essential to get a description of the context and background in this country: a. You say in the introduction that the Netherlands provides subsidies on the installation of solar panels? How much, when, where and how and under which conditions?”*

Response R3.5a: Thank you for this comment. To offer a more in-depth understanding of the local context, we have added a new Appendix B called “Solar energy in the Netherlands” to our manuscript. Therein, we now provide a detailed description of the importance of solar energy in the Netherlands, the role of households in contributing to the growth of solar energy, and the incentives available for households to install solar panels.

In our new Appendix B, we further discuss the subsidies provided by the Dutch government for the installation of solar panels. We outlined the specific details regarding the amount of subsidies available to households considering solar panels and highlighted the concept of net-metering. Furthermore, we elaborate on the tax impact of producing green energy with solar panels for Dutch households and thus provide local context for our field experiment.

Overall, we are confident that the new materials from above offer a rich background for readers to interpret our findings in the local context.

Comment R3.5b: *“b. How is the money saved/ earned done? Do owners of solar panels safe money because they use their own electricity, or can they sell back to the grid? Do energy taxes play a role? What are the implications of the current energy crisis and costs on the results?”*

Response R3.5b: Thank you for raising these questions. We have thus added a new Appendix B called “Solar energy in the Netherlands” to our manuscript.

In our new Appendix B, we describe how owners benefit from net-metering when they install solar panels. Net-metering allows households to obtain credit for supplying energy to the power grid that can later be used to offset the costs of energy consumption from the grid in times when a household's solar panels do not produce enough energy. Hence, households can significantly lower their energy bill by installing solar panels. Households may further apply for subsidies that support the installation of solar panels and profit from lower energy taxes due to the lower energy consumption from the grid.

We also discuss in our new Appendix B how the current energy crisis affects solar energy in the Netherlands. The high energy prices improved the business case for residential solar panels and thus led to additional solar panel installations by private households. Note, however, that our field experiment took place before the Russian invasion and thus before the energy crisis. Hence, it can be expected that the current savings from solar panels may be even larger.

Comment R3.5c: *“c. I don’t understand what the €813 saved/ earned refers to: is this saved in terms of electricity use? Does it consider installation costs?”*

Response R3.5c: Thank you for giving us the opportunity for clarification. The price effect of EUR 813 was set by the online retailer and is a conservative estimate of the potential savings that should hold for the majority of solar panels installed in the Netherlands, regardless of the specific location and other housing characteristics. The EUR 813 includes installation and maintenance costs as well as cost savings from reduced electricity use over an amortization period of 25 years. To address your comment, we have included the above explanation to our Method section. We further added more background information in our revised section “Presentation of interventions (field experiment)” in our Appendix. Therein, we explain that a newly installed solar energy system that covers the average electricity consumption of 2.81 MWh (Centraal Bureau voor de Statistiek 2023b) of a Dutch household in 2021 requires an initial investment of approximately EUR 4,000 (TU Delft 2023). The system incurs yearly operating costs of around 2% (Wirth 2023) of the initial investment, resulting in a total cost of EUR 5,920 over 25 years. Assuming that this system saves EUR 1,049.73 per year, which is the average amount spent on electricity by a household in the Netherlands in 2021, the total amount saved over 25 years would be EUR 20,323.15 or EUR 813 per year.

Comment R3.6: *“I regret that I don’t have a more positive message, but I hope that this feedback is useful. I wish you all the best for the future.”*

Response R3.6: Thank you very much for the detailed and constructive feedback. We followed all of your points carefully to bring our work to higher quality. We hope you agree.

References

- Amatulli, Cesare; Angelis, Matteo de; Peluso, Alessandro M.; Soscia, Isabella; Guido, Gianluigi (2019): The effect of negative message framing on green consumption: An investigation of the role of shame. In *J Bus Ethics* 157 (4), pp. 1111–1132.
- Ameli, Nadia; Brandt, Nicola (2015): What impedes household investment in energy efficiency and renewable energy? In *International Review of Environmental and Resource Economics* 8 (1), pp. 101–138.
- Berger, Sebastian; Kilchenmann, Andreas; Lenz, Oliver; Ockenfels, Axel; Schlöder, Francisco; Wyss, Annika M. (2022): Large but diminishing effects of climate action nudges under rising costs. In *Nature Human Behaviour* 6, pp. 1381–1385.
- Bleys, Brent; Defloor, Bart; van Ootegem, Luc; Verhofstadt, Elsy (2018): The environmental impact of individual behavior: Self-assessment versus the ecological footprint. In *Environment and Behavior* 50 (2), pp. 187–212.
- Bolderdijk, J. W.; Steg, L.; Geller, E. S.; Lehman, P. K.; Postmes, T. (2013): Comparing the effectiveness of monetary versus moral motives in environmental campaigning. In *Nature Climate Change* 3 (4), pp. 413–416.
- Camilleri, Adrian R.; Larrick, Richard P.; Hossain, Shajuti; Patino-Echeverri, Dalia (2019): Consumers underestimate the emissions associated with food but are aided by labels. In *Nature Climate Change* 9 (1), pp. 53–58.
- Carfora, V.; Catellani, P.; Caso, D.; Conner, M. (2019): How to reduce red and processed meat consumption by daily text messages targeting environment or health benefits. In *Journal of Environmental Psychology* 65, p. 101319.
- Centraal Bureau voor de Statistiek (2023a): Average energy prices for consumers. Available online at <https://www.cbs.nl/en-gb/figures/detail/84672ENG#>.
- Centraal Bureau voor de Statistiek (2023b): Energy consumption private dwellings; type of dwelling and regions. Available online at <https://www.cbs.nl/en-gb/figures/detail/81528ENG?q=parts%20of%20the%20country>.
- Dechezleprêtre, Antoine; Fabre, Adrien; Kruse, Tobias; Planterose, Blueberry; Sanchez, Ana; Stantcheva, Chico; Stantcheva, Stefanie (2022): Fighting climate change: International attitudes toward climate policies. Available online at https://www.oecd-ilibrary.org/economics/fighting-climate-change-international-attitudes-toward-climate-policies_3406f29a-en, updated on 2022.
- European Commission (2012): Europeans and their languages. Available online at <https://europa.eu/eurobarometer/surveys/detail/1049>.
- Evans, Laurel; Maio, Gregory R.; Corner, Adam; Hodgetts, Carl J.; Ahmed, Sameera; Hahn, Ulrike (2013): Self-interest and pro-environmental behaviour. In *Nature Climate Change* 3 (2), pp. 122–125.
- Ghesla, Claus; Grieder, Manuel; Schmitz, Jan; Stadelmann, Marcel (2020): Pro-environmental incentives and loss aversion: A field experiment on electricity saving behavior. In *Energy Policy* 137, p. 111131.
- Gillingham, Kenneth; Newell, Richard G.; Palmer, Karen (2009): Energy efficiency economics and policy. In *Annu. Rev. Resour. Econ.* 1 (1), pp. 597–620.
- Grazzini, Laura; Rodrigo, Padmali; Aiello, Gaetano; Viglia, Giampaolo (2018): Loss or gain? The role of message framing in hotel guests' recycling behaviour. In *Journal of Sustainable Tourism* 26 (11), pp. 1944–1966.
- Gupta, Shruti; Ogden, Denise T. (2009): To buy or not to buy? A social dilemma perspective on green buying. In *Journal of Consumer Marketing* 26 (6), pp. 376–391.

- Gustafson, Abel; Goldberg, Matthew H.; Bergquist, Parrish; Lacroix, Karine; Rosenthal, Seth A.; Leiserowitz, Anthony (2022): The durable, bipartisan effects of emphasizing the cost savings of renewable energy. In *Nature Energy* 7, pp. 1023–1030.
- Herberz, Mario; Hahnel, Ulf J. J.; Brosch, Tobias (2022): Counteracting electric vehicle range concern with a scalable behavioural intervention. In *Nature Energy* 7 (6), pp. 503–510. DOI: 10.1038/s41560-022-01028-3.
- Homar, Aja Ropret; Cvelbar, Ljubica Knežević (2021): The effects of framing on environmental decisions: A systematic literature review. In *Ecological Economics* 183, p. 106950.
- Karp, David Gutierrez (1996): Values and their effect on pro-environmental behavior. In *Environment and Behavior* 28 (1), pp. 111–133. DOI: 10.1177/0013916596281006.
- Kastner, Ingo; Stern, Paul C. (2015): Examining the decision-making processes behind household energy investments: A review. In *Energy Research & Social Science* 10, pp. 72–89.
- Minton, Ann P.; Rose, Randall L. (1997): The Effects of Environmental Concern on Environmentally Friendly Consumer Behavior: An Exploratory Study. In *Journal of Business Research* 40 (1), pp. 37–48. DOI: 10.1016/S0148-2963(96)00209-3.
- Nabi, Robin L.; Gustafson, Abel; Jensen, Risa (2018): Framing climate change: Exploring the role of emotion in generating advocacy behavior. In *Science Communication* 40 (4), pp. 442–468.
- Ölander, Folke; Thøgersen, John (1995): Understanding of consumer behaviour as a prerequisite for environmental protection. In *Journal of Consumer Policy* 18 (4), pp. 345–385.
- Poortinga, Wouter; Steg, Linda; Vlek, Charles (2004): Values, environmental concern, and environmental behavior. In *Environment and Behavior* 36 (1), pp. 70–93.
- Poortinga, Wouter; Whitaker, Louise (2018): Promoting the use of reusable coffee cups through environmental messaging, the provision of alternatives and financial incentives. In *Sustainability* 10 (3), p. 873.
- Rai, Varun; Reeves, D. Cale; Margolis, Robert (2016): Overcoming barriers and uncertainties in the adoption of residential solar PV. In *Renewable Energy* 89, pp. 498–505.
- Schwartz, Daniel; Loewenstein, George; Agüero-Gaete, Loreto (2020): Encouraging pro-environmental behaviour through green identity labelling. In *Nat Sustain* 3 (9), pp. 746–752. DOI: 10.1038/s41893-020-0543-4.
- Schwirplies, Claudia; Ziegler, Andreas (2016): Offset carbon emissions or pay a price premium for avoiding them? A cross-country analysis of motives for climate protection activities. In *Applied Economics* 48 (9), pp. 746–758.
- Segev, Sigal; Fernandes, Juliana; Wang, Weirui (2015): The Effects of gain versus loss message framing and point of reference on consumer responses to green advertising. In *Journal of Current Issues & Research in Advertising* 36 (1), pp. 35–51.
- Stern, Paul C.; Janda, Kathryn B.; Brown, Marilyn A.; Steg, Linda; Vine, Edward L.; Lutzenhiser, Loren (2016): Opportunities and insights for reducing fossil fuel consumption by households and organizations. In *Nature Energy* 1 (5), p. 16043.
- Stern, Paul C.; Wittenberg, Inga; Wolske, Kimberly S.; Kastner, Ingo (2018): Household production of photovoltaic energy. In Alan Lewis (Ed.): *The Cambridge Handbook of Psychology and Economic Behaviour*. Cambridge, UK: Cambridge University Press, pp. 541–566.
- Sussman, Reuven; Chikumbo, Maxine; Gifford, Robert (2018): Message framing for home energy efficiency upgrades. In *Energy and Buildings* 174, pp. 428–438.

TU Delft (2023): Detailed PV system design. Available online at <https://www.tudelft.nl/en/ewi/over-de-faculteit/afdelingen/electrical-sustainable-energy/photovoltaic-materials-and-devices/dutch-pv-portal/tools-models/detailed-pv-system-design>.

United States Environmental Protection Agency (2021): Greenhouse gas equivalencies calculator. Available online at <https://www.epa.gov/energy/greenhouse-gas-equivalencies-calculator>.

van der Linden, Sander (2018): Warm glow is associated with low- but not high-cost sustainable behaviour. In *Nature Sustainability* 1 (1), pp. 28–30.

White, Katherine; Habib, Rishad; Hardisty, David J. (2019): How to SHIFT Consumer Behaviors to be More Sustainable: A Literature Review and Guiding Framework. In *Journal of Marketing* 83 (3), pp. 22–49.

Wirth, Harry (2023): Recent facts about photovoltaics in Germany. Fraunhofer ISE. Available online at <https://www.ise.fraunhofer.de/en/publications/studies/recent-facts-about-pv-in-germany.html>.

REVIEWER COMMENTS

Reviewer #1 (Remarks to the Author):

I'm satisfied with the authors' responses to my comments and find the manuscript much improved. I found a few minor issues that still need attention though.

P. 13-14, l. 231-235, formulations are unclear and give an exaggerated impression of how surprised one should be of the bigger effect of the personal than the environmental framing. As now stated clearly other places in the manuscript, it is no surprise that the economic cost-benefit calculation is important when households consider making big investments, such as in solar panels. As is also now made clear other places, environmental motivations are likely one of the reasons why a household consider investing in solar panels and therefore visit the website. What we know from the experiment is that, when providing additional information to what's already on the website, in a pop-up window, additional concrete information about environmental savings is more effective than abstract environmental information. However, it is not correct that you: "do not find evidence that messages that target the environment are effective." The absolute effect of environmental information is not estimated, which would have to account also for the environmental information already on the website. I suggest re-formulation of lines 231-235 along the following lines: "However, contrary to prior research, we do not find evidence that messages that emphasize environmental benefits are effective. While the environmental aspect of solar panels may serve as an important initial motivator for customers to visit the website, it appears that the economic risks and potential gains associated with solar panels are essential for taking the last steps and making a commitment to adopt solar panels."

P. 15, l. 261: Change "our" to "a supplementary".

P. 39, l. 797: Change "diver" to "driver".

P. 59: The coefficients and confidence intervals do not fit the interpretation, or Figure S5. Most of the coefficients are non-significant and if this is correct it makes no sense to claim some as more effective than the others. Also, the few that are significant don't fit with the field experiment. Please double-check, correct errors, and clarify.

Reviewer #2 (Remarks to the Author):

In my judgment, the revised paper should be tentatively accepted for publication, pending several small changes. Many of these are editorial, but I think some changes in terminology and presentation are also warranted.

I suggest that somewhat different word usages might increase precision. In some places, I think "solar panels" should be recognized to mean "photovoltaic energy systems" or "photovoltaic energy production". Also, I think the authors should say more prominently that the paper is about commitment/adoption among retail consumers. For example, the second sentence of the abstract might read: "Hence, adoption by retail consumers has been a key barrier." I think such clarifications, though mainly editorial, might be advisable for clarity now and accessibility for future literature reviews.

I saw redundancy in language in a few places in the revised paper. The most striking was in the discussion section, which seemed to repeat language from the introduction. I think some shortening of the manuscript can be achieved here and encourage the authors to do that.

I suggest that the actual wordings of the four messages that constitute the independent variable in the experiment be presented, perhaps in the methods section, both for greater clarity and to allow readers to judge for themselves how much they agree with the authors' characterizations of the messages' main differences. This might add to length, but cutting redundant text can compensate.

In the discussion, I suggest mentioning any regulatory or financial incentives the Netherlands provides for solar adoption, as the findings here are likely contingent on the country's energy policy and economic contexts. Results might differ, for example, with stronger (or absent) incentives or regulations.

I have some difficulties with Figure 1, as it suggests that this study measures "actual behavior", but if the actual behavior is adoption of solar panels, it does not measure this behavior. I haven't checked the other references cited in the "actual behavior" column to see which ones measured actual behavior. Laboratory studies typically measure actual (not self-reported) behavior, but only behavior in a simulated context. So, equating lab with self-report and field with actual behavior seems misleading to me. I think the figure deserves some rethinking and redesign. Also, much of the figure caption is redundant with the text, so is not needed in the caption.

As I read the results, they could be encapsulated thus: "All four messages increased the rate of commitment to adopting solar panels significantly compared with the baseline. However, the framing in terms of financial savings for self was by far the most effective, producing about a 40% higher level of commitment than the baseline and about 27.5% higher than the average of the other three framings, which were not significantly different in effect from each other." (My numbers are only approximate.) I think a summary statement like this, if presented in the abstract, will both be accurate and draw attention to the paper. A suggested revision to the abstract is presented below for the authors' consideration.

I think the study's location in the Netherlands should appear in the abstract. A qualification that the findings may be specific to populations facing regulations and incentives similar to those in the Netherlands should also be made prominent in the paper's discussion section.

The section on environmental impact seems to base its estimates on the presumption that all the consumers who make a serious commitment to solar panels actually follow through and install them. Thus, some qualification is warranted here. Also, what was conducted looks like an environmental benefit analysis rather than a cost-benefit analysis, as costs did not come into this analysis.

I believe these changes can be accomplished without sending a further revised manuscript out for re-review.

Suggested revised abstract:

Green technologies such as solar panels foster the use of clean energy, yet often involve large-scale investments. Hence, adoption by retail consumers has been a key barrier. In this paper, we conducted a large-scale randomized controlled trial with a nationwide online retailer in the Netherlands (N = 26,873 participants) and provide empirical evidence in the field that message framing can significantly increase customers' serious commitment to adopting solar panels. We designed four messages aimed at promoting the purchase behavior of solar panel installations. Our messages presented outcomes for oneself or for the environment and highlighted cost savings or earnings (for oneself) or reducing emissions or generating green electricity (for the environment). All four messages significantly increased the rate of commitment to adopting solar panels compared with the baseline. However, the framing in terms of financial savings for self was by far the most effective, producing about a 40% higher level of commitment than the baseline and about 27.5% higher than the average of the other three framings, which were not significantly different in effect from each other.

Revised submission to *Nature Communications*

Manuscript: "Message framing to promote solar panels" (NCOMMS-23-05489)

Dear reviewers,

We sincerely appreciate the time and effort you have invested in assessing our work, and we would like to express our gratitude for your constructive comments and suggestions. Based on your feedback, we have revised our manuscript closely following your comments and suggestions.

We provide point-by-point responses below. To facilitate the review, we highlighted key changes in our manuscript in **red text color**.

RESPONSES TO REVIEWER #1

Comment R1.0: *“I’m satisfied with the authors’ responses to my comments and find the manuscript much improved. I found a few minor issues that still need attention though.”*

Response R1.0: Thank you for your positive feedback and for acknowledging the improvements in our manuscript. We appreciate your diligence in reviewing the revised version of our manuscript and have closely followed your feedback to address the remaining issues you have identified.

Comment R1.1: *“P. 13-14, l. 231-235, formulations are unclear and give an exaggerated impression of how surprised one should be of the bigger effect of the personal than the environmental framing. As now stated clearly other places in the manuscript, it is no surprise that the economic cost-benefit calculation is important when households consider making big investments, such as in solar panels. As is also now made clear other places, environmental motivations are likely one of the reasons why a household consider investing in solar panels and therefore visit the website. What we know from the experiment is that, when providing additional information to what’s already on the website, in a pop-up window, additional concrete information about environmental savings is more effective than abstract environmental information. However, it is not correct that you: “do not find evidence that messages that target the environment are effective.” The absolute effect of environmental information is not estimated, which would have to account also for the environmental information already on the website. I suggest re-formulation of lines 231-235 along the following lines: “However, contrary to prior research, we do not find evidence that messages that emphasize environmental benefits are effective. While the environmental aspect of solar panels may serve as an important initial motivator for customers to visit the website, it appears that the economic risks and potential gains associated with solar panels are essential for taking the last steps and making a commitment to adopt solar panels.””*

Response R1.1: Thank you for this comment. We appreciate the feedback and have re-formulated our discussion following your excellent suggestion.

Comment R1.2: *“P. 15, l. 261: Change “our” to “a supplementary”.”*

Response R1.2: Thank you. We fixed this.

Comment R1.3: *“P. 39, l. 797: Change “diver” to “driver”.”*

Response R1.3: Thank you. We fixed this.

Comment R1.4: *“P. 59: The coefficients and confidence intervals do not fit the interpretation, or Figure S5. Most of the coefficients are non-significant and if this is correct it makes no sense to claim some as more effective than the others. Also, the few that are significant don’t fit with the field experiment. Please double-check, correct errors, and clarify.”*

Response R1.4: Thank you for your feedback regarding the supplementary experiment. We appreciate your attention to detail, especially around our interpretation of coefficients. We have carefully reviewed the data

and analysis along your feedback and revised our interpretation. We have also revised the caption of our Figure S5b to clarify that we show the estimated rate of customer commitments and 95% confidence intervals (i.e., after applying the exponential transformation to the original coefficients). For additional transparency, we have also added a new Table S8 to the supplementary materials, where we provide detailed regression results for the lab experiment.

RESPONSES TO REVIEWER #2

Comment R2.0: *“In my judgment, the revised paper should be tentatively accepted for publication, pending several small changes. Many of these are editorial, but I think some changes in terminology and presentation are also warranted.”*

Response R2.0: Thank you for your positive feedback and detailed suggestions on the revised version of our paper. We followed your suggestions closely which has greatly improved our submission.

Comment R2.1: *“I suggest that somewhat different word usages might increase precision. In some places, I think “solar panels” should be recognized to mean “photovoltaic energy systems” or “photovoltaic energy production”. Also, I think the authors should say more prominently that the paper is about commitment/adoption among retail consumers. For example, the second sentence of the abstract might read: “Hence, adoption by retail consumers has been a key barrier.” I think such clarifications, though mainly editorial, might be advisable for clarity now and accessibility for future literature reviews.”*

Response R2.1: Thank you for this excellent suggestion. We now explicitly state that we refer to “photovoltaic systems.” Additionally, we have revised our abstract and manuscript to emphasize our focus on retail consumers.

Comment R2.2: *“I saw redundancy in language in a few places in the revised paper. The most striking was in the discussion section, which seemed to repeat language from the introduction. I think some shortening of the manuscript can be achieved here and encourage the authors to do that.”*

Response R2.2: Thank you for your comment. We have revised our manuscript to reduce redundancies and improve clarity and coherence, especially in the discussion section.

Comment R2.3: *“I suggest that the actual wordings of the four messages that constitute the independent variable in the experiment be presented, perhaps in the methods section, both for greater clarity and to allow readers to judge for themselves how much they agree with the authors’ characterizations of the messages’ main differences. This might add to length, but cutting redundant text can compensate.”*

Response R2.3: Thank you for this excellent comment. Following your suggestion, we have revised our methods section and added our four messages.

Comment R2.4: *“In the discussion, I suggest mentioning any regulatory or financial incentives the Netherlands provides for solar adoption, as the findings here are likely contingent on the country’s energy policy and economic contexts. Results might differ, for example, with stronger (or absent) incentives or regulations.”*

Response R2.4: Thank you for this comment. We now spell out clearly in different parts of the paper that our focus is on the Netherlands (e.g., in the abstract).

Comment R2.5: *“I have some difficulties with Figure 1, as it suggests that this study measures “actual behavior”, but if the actual behavior is adoption of solar panels, it does not measure this behavior. I haven’t checked the other references cited in the “actual behavior” column to see which ones measured actual behavior. Laboratory studies typically measure actual (not self-reported) behavior, but only behavior in a simulated context. So, equating lab with self-report and field with actual behavior seems misleading to me. I think the figure deserves some rethinking and redesign. Also, much of the figure caption is redundant with the text, so is not needed in the caption.”*

Response R2.5: Thank you for your comment. We revised Figure 1 along your suggestions. Additionally, we have edited the caption to eliminate redundancies with our main text.

Comment R2.6: *“As I read the results, they could be encapsulated thus: “All four messages increased the rate of commitment to adopting solar panels significantly compared with the baseline. However, the framing in terms of financial savings for self was by far the most effective, producing about a 40% higher level of commitment than the baseline and about 27.5% higher than the average of the other three framings, which were not significantly different in effect from each other.” (My numbers are only approximate.) I think a summary statement like this, if presented in the abstract, will both be accurate and draw attention to the paper. A suggested revision to the abstract is presented below for the authors’ consideration.”*

Response R2.6: Thank you for this excellent idea. Following your suggestion we have revised our abstract and incorporated a summary statement of our results.

Comment R2.7: *“I think the study’s location in the Netherlands should appear in the abstract. A qualification that the findings may be specific to populations facing regulations and incentives similar to those in the Netherlands should also be made prominent in the paper’s discussion section.”*

Response R2.7: Thank you for this comment. We agree with your feedback and have revised our Abstract accordingly. We further revised our discussion section to acknowledge that our findings may be specific to the regulatory framework in the Netherlands.

Comment R2.8: *“The section on environmental impact seems to base its estimates on the presumption that all the consumers who make a serious commitment to solar panels actually follow through and install them. Thus, some qualification is warranted here. Also, what was conducted looks like an environmental benefit analysis rather than a cost-benefit analysis, as costs did not come into this analysis.”*

Response R2.8: Thank you for this feedback which helped us clarify our potential impact section. In the revised version of our potential impact section, we have added qualifications and explicitly stated our presumption that all customers follow through on their commitment to solar panels. We have further changed our wording from cost-benefit analysis to environmental-benefit analysis.

Comment R2.9: *“I believe these changes can be accomplished without sending a further revised manuscript out for re-review.*

Suggested revised abstract:

Green technologies such as solar panels foster the use of clean energy, yet often involve large-scale investments. Hence, adoption by retail consumers has been a key barrier. In this paper, we conducted a large-scale randomized controlled trial with a nationwide online retailer in the Netherlands (N = 26,873 participants) and provide empirical evidence in the field that message framing can significantly increase customers' serious commitment to adopting solar panels. We designed four messages aimed at promoting the purchase behavior of solar panel installations. Our messages presented outcomes for oneself or for the environment and highlighted cost savings or earnings (for oneself) or reducing emissions or generating green electricity (for the environment). All four messages significantly increased the rate of commitment to adopting solar panels compared with the baseline. However, the framing in terms of financial savings for self was by far the most effective, producing about a 40% higher level of commitment than the baseline and about 27.5% higher than the average of the other three framings, which were not significantly different in effect from each other."

Response R2.9: Thank you for your detailed suggestion. We are glad to see that you do not see the need for another formal round of re-review. We have also adopted your suggested abstract.